# An integrated computational and experimental study uncovers FUT9 as a metabolic driver of colorectal cancer

Noam Auslander[1,*] (ID), Chelsea E Cunningham[2], Behzad M Toosi[2], Emily J McEwen[2], Keren Yizhak[3], Frederick S Vizeacoumar[2], Sreejit Parameswaran[2], Nir Gonen[4], Tanya Freywald[2], Kalpana K Bhanumathy[2], Andrew Freywald[2,**] (ID), Franco J Vizeacoumar[2,5,***] (ID) & Eytan Ruppin[1,****] (ID)

## Abstract

Metabolic alterations play an important role in cancer and yet, few metabolic cancer driver genes are known. Here we perform a combined genomic and metabolic modeling analysis searching for metabolic drivers of colorectal cancer. Our analysis predicts FUT9, which catalyzes the biosynthesis of Ley glycolipids, as a driver of advanced-stage colon cancer. Experimental testing reveals FUT9's complex dual role; while its knockdown enhances proliferation and migration in monolayers, it suppresses colon cancer cells expansion in tumorspheres and inhibits tumor development in a mouse xenograft models. These results suggest that FUT9's inhibition may attenuate tumor-initiating cells (TICs) that are known to dominate tumorspheres and early tumor growth, but promote bulk tumor cells. In agreement, we find that FUT9 silencing decreases the expression of the colorectal cancer TIC marker CD44 and the level of the OCT4 transcription factor, which is known to support cancer stemness. Beyond its current application, this work presents a novel genomic and metabolic modeling computational approach that can facilitate the systematic discovery of metabolic driver genes in other types of cancer.

**Keywords** colon cancer; FUT9; genome-scale metabolic modeling; oncogene; tumor suppressor
**Subject Categories** Cancer; Genome-Scale & Integrative Biology Metabolism
**Mol Syst Biol. (2017) 13: 956**

## Introduction

The initiation and development of cancer is known to be associated with major metabolic alterations (Hanahan & Weinberg, 2011; Ward & Thompson, 2012), leading to the recognition of transformed metabolism as one of the cancer hallmarks (Hanahan & Weinberg, 2011). Several metabolic abnormalities are quite general, including a shift in glucose metabolism from oxidative phosphorylation to aerobic glycolysis termed the Warburg effect, which is accompanied by lactate production and increased glucose uptake (Hsu & Sabatini, 2008). Other metabolic alterations are more tumor specific; different tumors differ in their dependence on glutamine (Son et al, 2013), serine (Possemato et al, 2011), or TCA cycle function (Selak et al, 2005; Dang et al, 2010). Yet, only few metabolic genes are presently known to be directly implicated in tumorigenesis. Those include mutations/loss of the genes encoding succinate dehydrogenase (SDH) complex subunits, which may cause paraganglioma (Frezza et al, 2011), the inactivation and loss of fumarate hydratase (FH), playing a casual role in hereditary leiomyomatosis and renal cell cancer (HLRCC) (Kiuru et al, 2002), and mutations in IDH1 and IDH2, which can lead to low-grade gliomas and acute myeloid leukemia (AML) (Parsons et al, 2008; Dang et al, 2009; Mardis et al, 2009; Sciacovelli et al, 2016; Sykes et al, 2016). Overall, there is still much more to learn about the causal role of metabolic genes in cancer.

Here we take a genome-wide computational approach to identify metabolic genes that may cause a tumorigenic transformation. We focus on colorectal cancer, which is initiated by a polyp that grows from the mucosa and becomes cancerous at some point. To this end, we performed a two-step computational analysis, which is based first on a molecular analysis of patient tumors followed by a genome-scale

1 Department of Computer Science, Center for Bioinformatics and Computational Biology, University of Maryland, College Park, MD, USA
2 Department of Pathology, Cancer Cluster, College of Medicine, University of Saskatchewan, Saskatoon, SK, Canada
3 Broad Institute of Harvard and MIT, Cambridge, MA, USA
4 Sagol School of Neuroscience, Tel-Aviv University, Tel-Aviv, Israel
5 Cancer Research, Saskatchewan Cancer Agency, Saskatoon, SK, Canada
*Corresponding author. Tel: +1 301 405 5936; E-mail: noamaus@gmail.com
**Corresponding author. Tel: +1 306 966 5248; E-mail: andrew.freywald@usask.ca
***Corresponding author. Tel: +1 306 966 7010; E-mail: franco.vizeacoumar@usask.ca
****Corresponding author. Tel: +1 301 405 5936; E-mail: eyruppin@gmail.com

metabolic modeling (GSMM) analysis in the second step. A genome-scale metabolic model (GSMM) is a computer program built around a set of reactions that comprise a metabolic network, accompanied by a mapping of genes and proteins to the reactions they catalyze within the network (Orth et al, 2010). GSMM of human metabolism has become feasible in recent years thanks to the publication of the first full-fledged genome-scale human metabolic models [Recon1 (Duarte et al, 2007; Ma et al, 2007)]. In addition to a network of more than 3000 metabolic reactions, Recon1 contains Boolean mappings of approximately 1,500 metabolic genes through their encoded enzymes to these reactions, sub-cellular compartmentalization of processes and pathways, and manually curated reaction stoichiometry and membrane transporters. A key critical merit of GSMM modeling is that it does not require the explication of detailed enzymatic kinetic information (which is yet unknown on a network scale) as it describes the metabolic state of cells at steady state. GSMM enables the integration of omics data collected at specific conditions to provide a genome-wide view of their corresponding metabolism, that is, the prediction of the likely metabolic fluxes across the network, including uptake and secretion rates, cell proliferation, and more. GSMMs can also be used to predict the phenotypic effects of genetic and environmental perturbations on the cell's flux distribution and viability. Such modeling studies have been employed in recent years to describe human metabolism (Duarte et al, 2007) in general and in cancer (Folger et al, 2011; Agren et al, 2012, 2014; Nam et al, 2014; Yizhak et al, 2014).

Our analysis identifies the FUT9 gene, encoding alpha-(1,3)-fucosyltransferase, as the top predicted metabolic tumor suppressor in colorectal cancer. Our subsequent experimental study of FUT9 function indicates that it plays a more complex, dual role in this malignancy; its expression in TICs favors tumor initiation, while subsequent colorectal cancer progression via the mass of colon cancer bulk tumors is supported by its downregulation.

# Results

## An integrated genomic modeling analysis predicts a causal complex role of FUT9 in driving colon cancer

We developed a two-step computational approach to predict metabolic tumor suppressors, that is, genes whose downregulation promotes cancer. Applied to study colon cancer, the first step employs a straightforward genomic analysis of the Cancer Genome Atlas (TCGA) database (Beroukhim et al, 2010; Barretina et al, 2012) to identify metabolic genes that are downregulated in colorectal cancer (Fig 1A). Subsequently, we performed a novel metabolic modeling analysis to identify, among the genes identified as associated with tumorigenesis in the first step, those whose downregulation is indeed most likely to result in the metabolic alterations observed in colorectal tumors and thus are more likely to play an actual causal role in the transformation of normal to cancerous tissues (Fig 1B). A detailed overview of each step follows.

### Genomic identification of 34 candidate metabolic tumor suppressor genes in colorectal cancer

This step consists of three sub-steps that are applied sequentially, analyzing gene expression, copy number (CN), and survival data

from 272 colorectal cancer samples and 42 matching healthy colon tissues samples in the TCGA (Beroukhim et al, 2010; Barretina et al, 2012): (i) First, analyzing the transcriptomic data of these samples, we identified 4593 genes that are significantly downregulated in colon cancer (one-sided Wilcoxon rank-sum test with multiple hypothesis correction ($\alpha$ = 0.001), Table EV1). (ii) Second, 328 of these downregulated genes have significantly lower copy number in the tumors compared to the healthy samples ($Q$-values < 0.25, Table EV2). (iii) Finally, a Kaplan–Meier survival analysis further narrowed down this list to 177 candidate tumor suppressors whose downregulation is negatively correlated with patient survival (and thus, likely to enhance tumor progression; see Materials and Methods, Fig 1A and Table EV3). Reassuringly, the resulting list includes several known colon tumor suppressors such as APC (Fearnhead et al, 2001; Aoki & Taketo, 2007), TCF7L2 (Hazra et al, 2008; Slattery et al, 2008), MCC (Kinzler et al, 1991), PTEN (Nassif et al, 2004; Song et al, 2012), and SMAD4 (Miyaki et al, 1999; Alazzouzi et al, 2005). It also includes 34 metabolic genes that are present in the human metabolic model (Table EV4), and which we further studied in the next modeling step.

### A GSMM analysis points to FUT9 as the top predicted metabolic tumor suppressor gene in colorectal cancer

To predict metabolic genes whose downregulation may play a causal role in colorectal cancer, we utilized a GSMM analysis approach termed the Metabolic Transformation Algorithm (MTA) (Yizhak et al, 2013). This algorithm was previously developed and used to successfully identify life-extending metabolic genes in yeast (Yizhak et al, 2013) and is employed here for the first time to search for metabolic tumor suppressors in cancer. MTA is a generic algorithm that aims to identify metabolic gene knockouts that are capable of driving a transformation from a given metabolic state to another, defined target state. The inputs to MTA are the pertaining transcriptomic measurements of these two given and targets states. Its output is a ranked list of metabolic genes whose inactivation has the potential to induce the transformation from the given to the target states (Materials and Methods) (Yizhak et al, 2013). In our case, the given metabolic state is the healthy, non-malignant state, and the target state is the cancerous one, and correspondingly, the inputs to the algorithm are a set of gene expression data from matched healthy and tumor colon samples.

While the original publication of MTA has mainly focused on its testing and validation in a known collection of gene knockouts in microorganisms, it already showed that MTA correctly identifies fumarate hydratase as a gene whose knockdown may cause the metabolic transformations observed in HLRCC (Kiuru et al, 2002; King et al, 2006). We now tested and validated that MTA successfully identifies the knockdown of succinate dehydrogenase (SDH) as a likely cause of the metabolic alterations observed in hereditary paraganglioma (Frezza et al, 2011). To further test the ability of MTA to identify the genes that were knocked down in mammalian screens from the pertaining transcriptomic data, we further mined the literature to assemble a collection of 19 datasets of metabolic genes for which we found mouse or human gene expression data before and after the knockdown of each of these genes (Appendix, Table EV5). For each of these knockdowns, we gave MTA these transcriptomic data as inputs and applied it to predict the most likely genes whose knockdowns may account for the transcriptomic

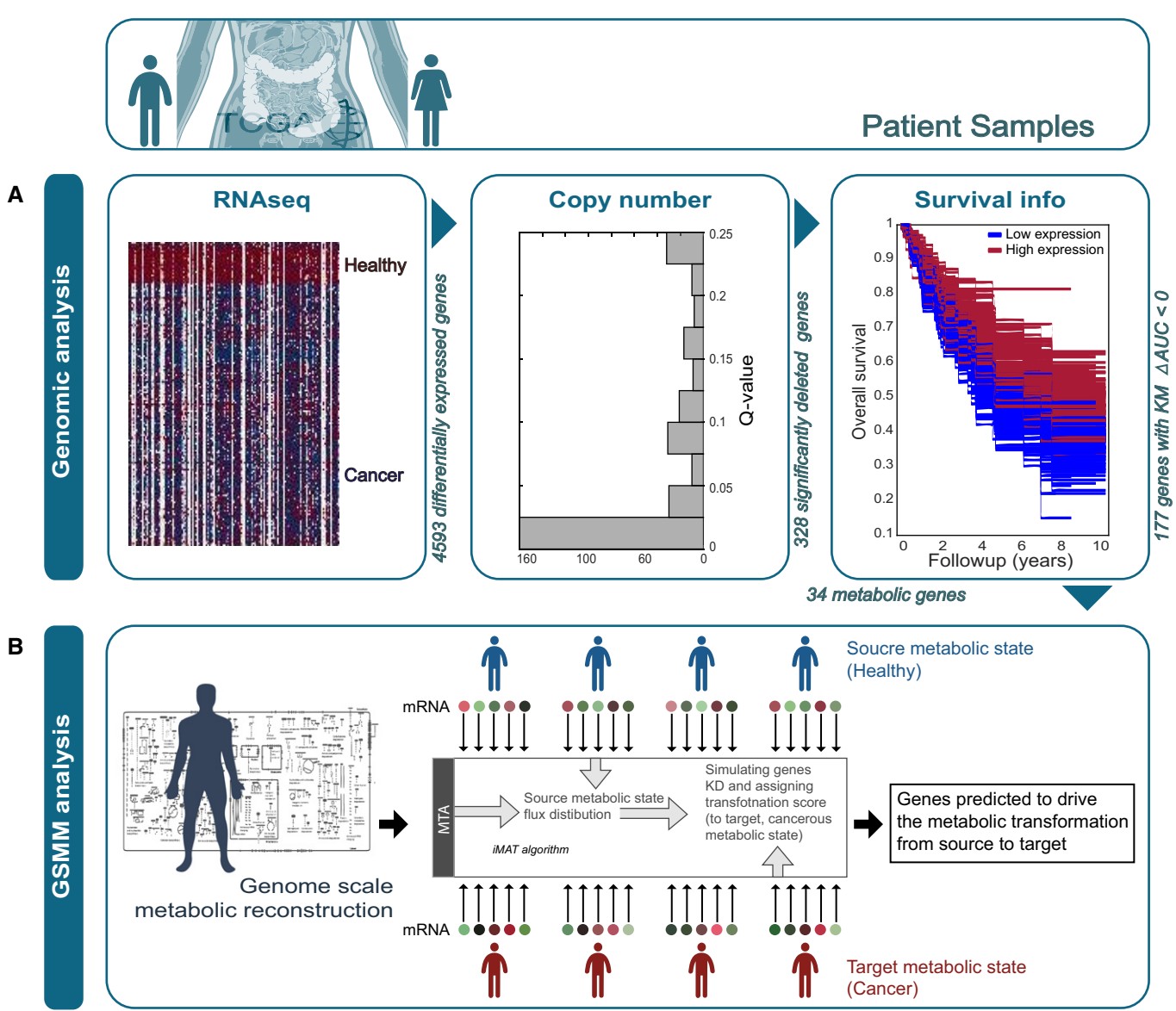

**Figure 1.  Two-step pipeline for predicting metabolic tumor suppressors.**

A  Genomic analysis of three types of data yields an initial list of potential tumor suppressors.

B  GSMM-based approach of the potential tumor suppressors identifies metabolic genes whose knockdown may play a causal role in tumorigenesis.

changes observed in these experiments. MTA correctly predicted the experimentally knocked down genes in 13 of the 19 cases studied in the top 20% of the predictions (binomial *P*-value = 5.8266e-06, and its performance remains robust at multiple threshold setting, Appendix), validating MTA's predictive ability in mammalian tissues (Table EV5).

We then turned to apply MTA to identify metabolic genes that, when downregulated, can transform a healthy tissue to a cancerous one. We analyzed three independent transcriptomic datasets including 27 paired healthy/tumor samples from TCGA, 17 paired healthy/tumor samples from Khamas *et al* (Khamas *et al*, 2012), and 32 paired healthy/adenoma samples from Sabates-Bellver *et al* (2007). In the first step, we ran an MTA analysis on each pair of matched

healthy and tumor gene expression samples, yielding a ranked list of genes according to their *oncogenic transformation scores* (*OTS*) (Materials and Methods). OTS scores denote the likelihood that a gene knockout in the healthy cells can transform their metabolic state to a cancerous one. Following that, in a second step, an aggregate OTS was assigned to each metabolic gene by considering its scores across all samples and then, in a third step we aggregated the OTS scores of each gene across all three datasets analyzed. We additionally analyzed colon polyp data from Sabates-Bellver *et al* (2007), which includes 32 matched healthy and polyp samples. These data enabled us to perform two complementary MTA analyses, one predicting metabolic genes whose knockdown may cause the transformation to the polyp state, and one predicting metabolic genes

**Table 1.  34 candidate metabolic tumor suppressor genes in colorectal cancer.**

| Gene | Σ healthy → cancer OTS score | Healthy → adenoma OTS score | Adenoma → cancer OTS score | Differential expression P-value | CN Q-value | KM ΔAUC |
|------|------|------|------|------|------|------|
| FUT9 | 8.54 | 3.02 | 2.99 | 5.06E-24 | 0.0356 | −0.120669976 |
| AKR7A2 | 6.91 | 4.55 | 0.06 | 2.15E-14 | 3.46E-05 | −0.198482955 |
| CAT | 5.78 | 0 | 0 | 5.76E-19 | 0.215 | −0.124211074 |
| PTEN | 4.91 | 0.09 | 2.67 | 2.08E-19 | 0.00494 | −0.009581467 |
| PIK3CD | 4.3 | 0 | 0.2 | 1.79E-11 | 0.00205 | −0.048812134 |
| FUCA1 | 4.07 | 0 | 0 | 1.27E-23 | 3.46E-05 | −0.114652506 |
| PLCE1 | 3.47 | 1.3 | 0 | 3.33E-23 | 0.0458 | −0.104694133 |
| STS | 2.86 | 0 | 0.1 | 1.49E-08 | 0.0136 | −0.080255382 |
| SDHB | 2.81 | 2.4 | 0 | 3.72E-16 | 3.46E-05 | −0.106186688 |
| MAN1C1 | 2.6 | 0 | 0.2 | 9.61E-12 | 3.78E-05 | −0.023167238 |
| MTHFR | 2.14 | 0.21 | 0 | 2.06E-14 | 0.00205 | −0.162335156 |
| PIGN | 2.1 | 0 | 0 | 1.41E-09 | 0.187 | −0.029340173 |
| FH | 2.03 | 1.73 | 1.2 | 3.27E-11 | 1.24E-68 | −0.033206093 |
| PLA2G2D | 1.66 | 0.9 | 0 | 3.48E-09 | 0.000147 | −0.010922122 |
| SLC18A2 | 1.48 | 0 | 0.73 | 3.27E-15 | 0.19 | −0.095652366 |
| LIPC | 1.22 | 0.9 | 0 | 1.26E-19 | 0.215 | −0.005134395 |
| CYP2C18 | 1.2 | 1.64 | 0 | 1.43E-12 | 0.09 | −0.048710276 |
| HMGCL | 1.2 | 1.09 | 0 | 1.40E-20 | 3.46E-05 | −0.118954367 |
| ACADS | 1.11 | 2.4 | 0.12 | 2.73E-24 | 0.102 | −0.065005732 |
| PANK4 | 1.02 | 2.11 | 1.2 | 8.29E-13 | 0.0341 | −0.044198756 |
| COX6B2 | 0.82 | 0.76 | 0.1 | 1.88E-11 | 0.0397 | −0.024199841 |
| PDE4D | 0.8 | 2 | 0 | 1.01E-17 | 0.00629 | −0.076789143 |
| ECHS1 | 0.71 | 1.2 | 0 | 5.59E-12 | 0.172 | −0.054802279 |
| INPP5A | 0.32 | 3 | 0 | 2.59E-22 | 0.0599 | −0.031649119 |
| ITPKA | 0.2 | 1.01 | 0.8 | 2.28E-18 | 0.172 | −0.077477859 |
| SLC25A4 | 0.2 | 0.2 | 0 | 4.23E-12 | 0.00872 | −0.227357666 |
| HS3ST5 | 0.11 | 0 | 0 | 1.28E-09 | 0.0676 | −0.03426555 |
| FECH | 0 | 1.53 | 0 | 4.80E-17 | 0.227 | −0.102888235 |
| ME2 | 0 | 1.88 | 0 | 6.93E-17 | 0.0273 | −0.083078298 |
| NADK | 0 | 0.96 | 0.01 | 2.27E-14 | 0.0815 | −0.172031059 |
| NDUFB8 | 0 | 1.5 | 0 | 2.10E-11 | 0.234 | −0.023126419 |
| NMNAT1 | 0 | 0.98 | 0 | 8.38E-19 | 0.0062 | −0.124354125 |
| PAFAH2 | 0 | 0.1 | 0 | 3.67E-21 | 3.47E-05 | −0.021012791 |
| PC | 0 | 2.76 | 0 | 1.44E-16 | 0.0341 | −0.214743894 |

For each metabolic predicted tumor suppressor, the table displays: (i) the OTS scores for the three transformations, and genomic properties (ii) differential expression P-value, (iii) copy number (CN) deletion Q-value (P-value that has been adjusted for the false discovery rate), and (iv) Kaplan–Meier survival ΔAUC.

whose inactivation may cause a further malignant transformation into colon cancer (Materials and Methods, Table EV4).

The distribution of the resulting OTS scores of the 34 metabolic genes examined via these MTA analyses is presented in Table 1. While all 34 genes present genomic patterns that associate them with a tumorigenic state (using expression, copy number, and survival data), only few are predicted by MTA to causally transform the metabolic healthy state to that of a cancerous one. As evident, only the knockdown of PTEN and FUT9 is predicted to transform the metabolic state of healthy cells as well as that of adenoma cells to that of colorectal tumors with high OTS scores (Materials and Methods). FUT9 is the most highly scored gene and is also strongly supported by the earlier genomic analysis: Its expression is strongly downregulated in colon cancer (Rank-sum P-value = 1e-22, Fig 2A), it is significantly deleted in colon cancer while not in other cancer types (Q-value = 0.0356, Fig 2B), its low expression is associated with poor survival in colon cancer (Kaplan–Meier (KM) ΔAUC = −0.1206, Fig 2C) (Table 1) [The resulting KM log-rank

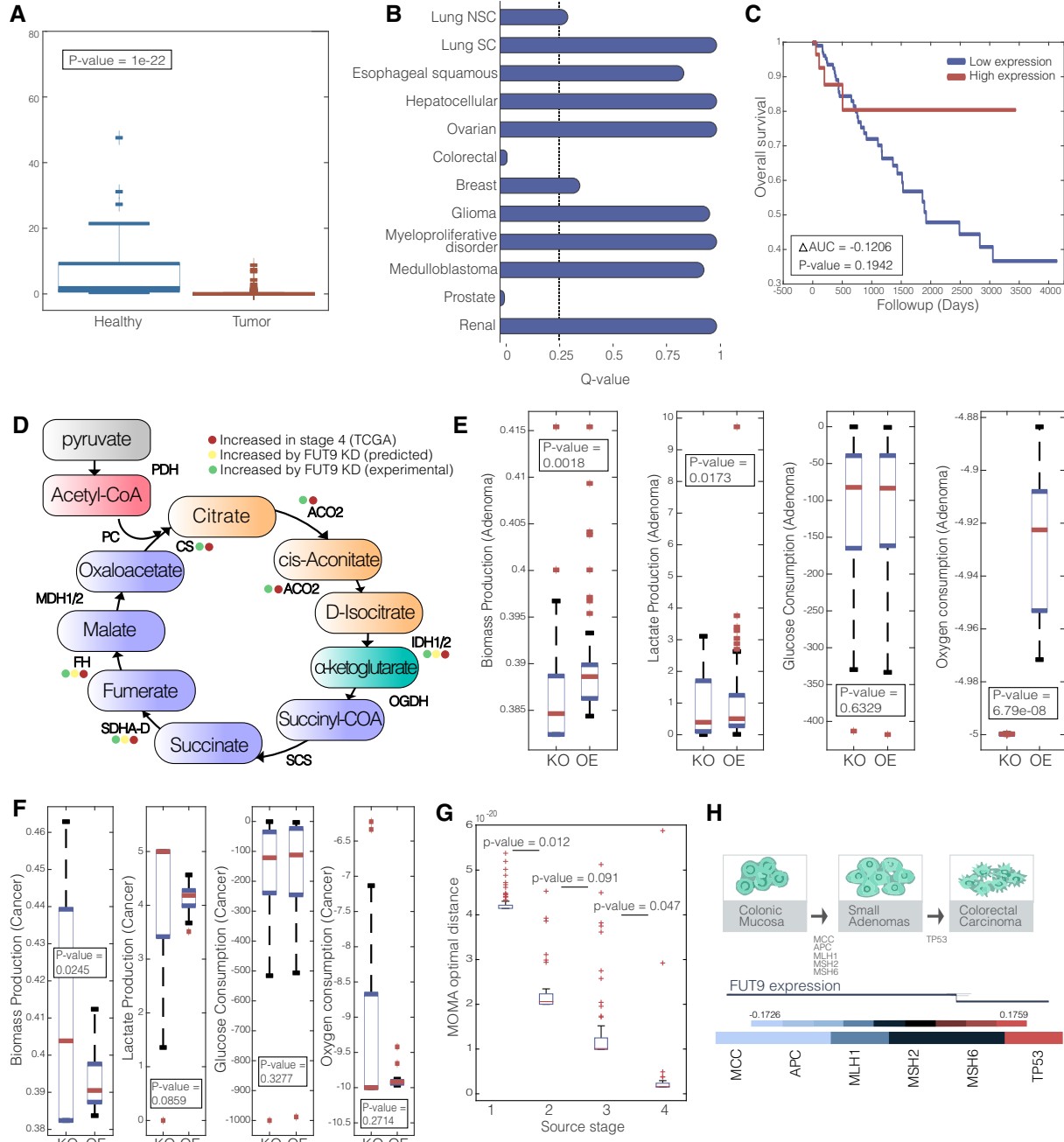

**Figure 2. Tumorigenic attributes of FUT9.**

A   A boxplot describing the expression of FUT9 in tumor vs. healthy colon tissues.

B   *Q*-value for CN of FUT9 in 12 different cancer types, the dashed line represents a significance threshold of 0.25.

C   Kaplan–Meier survival curve for FUT9 expression (top and bottom 0.5 quartiles).

D   The TCA cycle and its associated enzymes that are increased in stage 4 colorectal cancer (red), predicted to increase following FUT9 KD (yellow) and increase following FUT9 KD experimentally (green).

E   Boxplot showing the distribution of biomass production, glucose consumption, lactate production, and oxygen consumption in adenoma state when FUT9 is knocked down (KD) and overexpressed (OE).

F   Boxplot showing the distribution of biomass production, glucose consumption, lactate production, and oxygen consumption in cancer state when FUT9 is knocked down (KD) and overexpressed (OE).

G   Boxplots sowing the MOMA scores obtained by the knockdown of FUT9 in stages 1–4.

H   Upper panel: Colorectal adenoma–carcinoma sequence. Middle panel: the emerging role of FUT9 in colorectal tumor progression. Lower panel: Correlation heat map of FUT9 copy number (CN) and early- and late-stage prognostic markers of colorectal cancer.

Data information: (A, E–G) On each box, the central mark indicates the median, and the bottom and top edges of the box indicate the 25$^{th}$ and 75$^{th}$ percentiles, respectively. The *P*-values are for one-sided Wilcoxon rank-sum test.

P-value is 0.1942, likely due to the small sample size of patients expressing FUT9 (only ~15% of patients.)]. Interestingly though, while MTA highly scores FUT9 for all three transformations, FUT9 is not significantly downregulated at early-stage colon adenomas using paired gene expression of healthy/adenoma samples from Sabates-Bellver *et al* (2007) (Paired Student's *t*-test, P-value = 0.47, Appendix Fig S1). This suggests that its inactivation may play a significant role only at later stages of colon cancer progression. Bearing this observation in mind, we set to study the role of FUT9 further, first computationally and then experimentally.

## GSMM analysis of the metabolic implications of FUT9 inactivation

FUT9 belongs to the glycosyltransferase family and catalyzes the last step in the biosynthesis of Ley glycolipids in the carbohydrate antigen Lex (Nishihara *et al*, 2003; Gouveia *et al*, 2012). This reaction takes place in the Golgi compartment, and the product is transported to the cytosol and secreted out from the cell (Duarte *et al*, 2007). The Ley glycolipid was previously reported to inhibit the procoagulant activity and metastasis of human adenocarcinoma (Nudelman *et al*, 1986; Suzuki *et al*, 1997; Inufusa *et al*, 2001). The loss of FUT9 in the metabolic model prevents Ley glycolipid formation and secretion. To chart the network-wide metabolic alterations induced by FUT9 inactivation, we performed a Minimization Of Metabolic Adjustment (MOMA) (Segrè *et al*, 2002) analysis to predict the metabolic state after FUT9 KD in late-stage colorectal cancers, simulated by the Gene Inactivity Moderated by Metabolism and Expression (GIMME) algorithm (Becker & Palsson, 2008) (Materials and Methods). This pinpoints reactions whose flux is predicted to be most afflicted by FUT9 inactivation in advanced-stage cancer. We found that the loss of FUT9 in late-stage colorectal cancers is predicted to cause an increase in the flux of 25 reactions, and a decrease in the flux of six reactions (Table EV6). The flux is predicted to increase in reactions associated with glucose metabolism, and particularly TCA cycle (hyper-geometric P-value = 1.3676e-09, Fig 2D, Table EV6). We find that the expression of metabolic genes associated with reactions predicted to increase following FUT9 loss is significantly upregulated in stage 4 vs. stage 3 colon tumors when compared by their

expression in TCGA data (hyper-geometric P-value = 0.0046, Table EV6). Experimental evaluation of these predictions using the Human Glucose Metabolism, RT² Profiler™ PCR Array revealed a good correlation with our computational prediction (Fig 2D). In particular, 12 genes, including FH and SDHD, proved to be upregulated in FUT9-silenced cells as expected from our computational analyses (Appendix Fig S2).

To evaluate the effect of FUT9 knockdown (KD) and overexpression (OE) on biomass production, glucose consumption, lactate production, and oxygen consumption in the benign colon adenoma state, we (i) simulated the wild-type metabolic state associated with colon adenoma. This was done by incorporating adenoma gene expression data from Sabates-Bellver *et al* (2007) using the GIMME algorithm. (ii) We then sampled 100 flux distributions in the resulting predicted adenoma wild-type state. In each such sample, we applied the MOMA (Segrè *et al*, 2002) algorithm to predict the metabolic state after FUT9 KD and OE in adenoma, summing up the results overall 100 samples (Materials and Methods). We find that the biomass production predicted is significantly higher under FUT9 OE than its KD, as well as lactate secretion rate (Wilcoxon rank-sum P-value = 0.0081 and 0.0173, respectively, Fig 2E). While oxygen consumption rate is significantly higher under FUT9 KD (Wilcoxon rank-sum P-value = 6.79e-8, Fig 2E). These predictions imply that FUT9 activity is required for supporting cancer proliferation in the adenoma state, which are consistent with the genomic findings we reported above that, while FUT9 expression is strongly downregulated in colon cancer, it is not significantly downregulated at early-stage colon adenomas.

We next evaluated the metabolic effects of FUT KD and OE in the colon tumor state. To this end, we performed a similar analysis as described above for adenoma, while first inferring the likely metabolic state of colon tumors (Materials and Methods). Strikingly, we find that the predicted biomass production in the cancerous state is significantly higher under FUT9 KD than its OE (Wilcoxon rank-sum P-value = 0.0245, Fig 2F) and that lactate production rate is also increased under FUT9 KD (Wilcoxon rank-sum P-value = 0.0859, Fig 2F), opposite to the observed in simulated colon adenoma state. These predictions imply that the loss of FUT9, while hampering the growth of adenomas, is required for the proliferation of colon

---

**Figure 3.  Knockdown of FUT9 expression increases aggressiveness of colon cancer cells.**

A    HCT116 and DLD1 control and FUT9 knockdown cells were seeded evenly in 96-well plates, and the number of viable cells after 72 h was analyzed using Resazurin absorbance reading. The graph represents the mean ± s.e. from three independent replicates normalized to the control cells. Six wells per replicate were analyzed.

B    The same cells from (A) were seeded in soft agar and cultured for 28 days. The number of colonies formed was quantified relative to the control cells. The mean ± s.e. from two independent replicates are represented.

C    The fold change in gene expression for the FUT9 knockdown and FUT9-overexpressing cells were analyzed using RT–qPCR. The graphs represent the mean ± s.e. fold change from three independent replicates.

D, E    HCT116 and DLD1 FUT9 knockdown (D) or FUT9-overexpressing (E) cells were seeded at very low densities in a 24-well dish and cultured for 10 days. The number of colonies formed in each well was counted. The graph represents the mean ± s.e. of two independent replicates. Three wells were analyzed per replicate. Representative images of one well for each condition are shown.

F    HCT116 control and FUT9 knockdown cells were each seeded to form a confluent monolayer. A scratch was made in each monolayer and the width of the scratch monitored by imaging the same areas of each scratch (2 per scratch) at the time of scratching (0 h) and 24, 48, and 72 h later. The graph depicts the mean ± s.d. of two independent experiments and represents the percentage of scratch open at each time point relative to the 0-h point. For optimal presentation, individual scratch images are shown at different brightness and contrast settings. Scale bar: 400 μM.

G    The wound-healing assay was performed with HCT116 control and FUT9-overexpressing cells and analyzed as in (F). The graph summarizes the mean ± s.d. of two independent experiments and represents a percentage of scratch open at each time point relative to the 0-h point. For optimal presentation, individual scratch images are shown at different brightness and contrast settings. Scale bar: 400 μM.

Data information: *P < 0.05, Student's *t*-test.

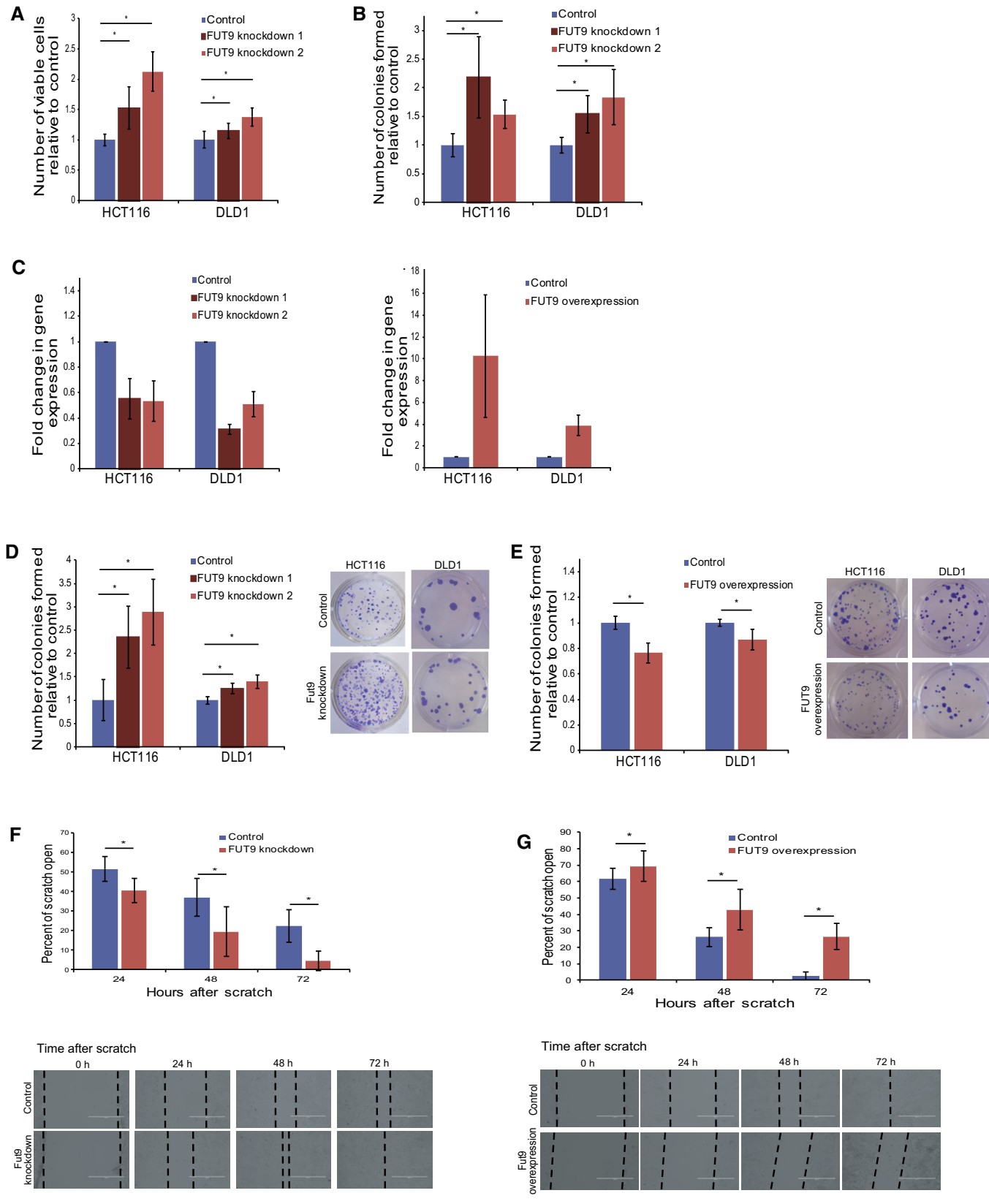

**Figure 3.**

tumors, while its overexpression significantly reduces proliferation in that state.

Given the opposite predicted effects of KD perturbation in colon adenomas vs. tumors, we performed an additional GSMM analysis to study whether FUT9 inactivation at early colorectal cancer stages can induce the metabolic state observed at advanced tumors, or only its inactivation at late stages can induce this transformation. To this end, we first inferred the likely metabolic state of advanced colorectal tumors using the GIMME algorithm (Becker & Palsson, 2008), as done above in the adenoma analysis. We then predicted the likely metabolic states after the loss if FUT9 in each of the four different stages of colorectal cancer progression, asking how similar is the metabolic state induced after the loss of FUT9 in each of these stages to the advanced, late cancerous state. The metabolic state after the KD of FUT9 in each stage-specific context was predicted using the MOMA algorithm (Segrè et al, 2002) (Materials and Methods). This analysis revealed that the loss of FUT9 at early stages does not bring the metabolic state close to that observed in advanced cancer. Rather, for the FUT9 loss to cause such an effect, it has to occur in later stages of the disease (Fig 2G). This indicates that FUT9 downregulation is a tumor-transformative event only if occurs at later stages of tumor progression. To study this further from a genomic perspective, we analyzed the correlation between FUT9 copy number and the copy number levels of known early and late genetic markers of colorectal cancer. We find that FUT9 expression levels negatively correlate with the loss of the early markers APC and MCC (Spearman $\rho = -0.1726$ and $-0.1707$, $P$-value < 0.05, respectively), while it is positively correlated with the loss of TP53, a marker of the advanced stage (Fearon, 1992; Lurje et al, 2007) (Spearman $\rho = 0.1759$, $P$-value < 0.05, Fig 2H).

## An experimental study of the predicted effects of FUT9 on driving colon cancer

### FUT9 knockdown increases colon cancer cells proliferation and migration

To test the prediction that FUT9 downregulation supports colon cancer aggressiveness, we silenced its expression using shRNA-based knockdowns in two colorectal cancer cell lines, HCT116 and DLD1 that express FUT9 (Fig 3C). Knockdown and control cells were seeded at equal numbers, allowed to propagate under normal conditions, and the abundance of viable cells was assessed by Resazurin

staining. These experiments revealed that the knockdown of FUT9 in both cell lines significantly increases their expansion compared to matching non-targeting shRNA controls (Fig 3A). In agreement, FUT9 silencing also enhanced anchorage-independent growth in soft agar (Mori et al, 2009) (Fig 3B). Moreover, the loss of FUT9 augmented the ability of cancer cells to form colonies in monolayers, when cells were seeded at very low densities (Fig 3D). To further assess the specificity of our observations, we constitutively overexpressed FUT9 under the CMV promoter (Fig 3C). Consistent with the effect of FUT9 silencing, FUT9 overexpression suppressed the colony formation capacity of both HCT116 and DLD1 cells (Fig 3E). The aggressiveness of tumor cells not only depends on their ability to proliferate, but also on their ability to migrate and invade surrounding tissues. To determine the effect of FUT9 on migration of colorectal cancer cells, we performed wound-healing assays (Liang et al, 2007). These experiments show that FUT9 knockdown enhances cell migration (Fig 3F), while FUT9 overexpression produces the opposite response (Fig 3G). Interestingly, changes in glycosylation pattern are known to modulate cell migration, cell–cell adhesion, cell signaling, growth, and metastasis (Hakomori, 1996; Fuster & Esko, 2005; de Freitas Junior & Morgado-Díaz, 2016). As FUT9 belongs to the glycosyltransferase family, to gain mechanistic insight we next investigated how loss of FUT9 may enhance cell migration. We used an RT$^2$ Profiler human glycosylation PCR array (Qiagen, 330231 PAHS-046ZA) to examine the changes in the expression pattern of glycosylation-related genes in FUT9 knockdown cells when compared to control cells (Appendix Fig S3). Interestingly, we found downregulation of the glucosidase II alpha subunit GANAB (fold change = −11.28) and Core2/Core4 beta-1,6-N-acetylglucosaminyltransferase GCNT3 (fold change = −3.55) (Appendix Fig S3A). Downregulation of both these genes has been previously shown to lead to enhanced cell migration and invasion, resulting in aggressive cancers, consistent with our results showing FUT9 knockdown leads to increased colony formation and cell migration (Fig 3D and F) (Huang et al, 2006; Chiu et al, 2011). Previous studies in colorectal cancer have shown that higher levels of core 1 glycans, T antigen, and Tn antigen are the most predominantly observed O-glycosylation changes (Holst et al, 2015). Consistent with this, loss of FUT9 showed an upregulation of the N-acetylgalactosaminyl transferases, GALNT8 (fold change = 11.80), GALNT13 (fold change = 4.21), and GALNT12 (fold change = 1.94) (Appendix Fig S3B). These three enzymes catalyze the formation of the Tn antigen by transferring

**Figure 4. Expression of FUT9 supports tumor development.**

A HCT116 FUT9 knockdown and matching control cells were seeded in ultra-low attachment plates and cultured for 1 week. The resulting tumorspheres were collected, dissociated, and the total number of cells counted. The graph represents the mean ± s.e. of two independent replicates normalized to the number of control cells. Each replicate represented tumorsphere cells collected from 24 independent wells. Representative images are shown. Scale bar, 1,000 μm.

B FUT9-overexpressing and control cells were cultured as tumorspheres and analyzed as in (A). Two independent replicates and representative pictures are depicted. The graph represents the mean ± s.e. of two independent replicates normalized to the number of control cells. Representative images are shown. Scale bar, 1,000 μm.

C CD44 expression in FUT9 knockdowns (in red) and shRFP control (in blue) in HCT116 cells was assessed using anti-CD44 and flow cytometry, and representative histograms were overlayed (second panel). Isotype controls were also plotted and overlayed (first panel). Median fluorescent intensity (MFI) values derived from the software are plotted as bar chart. The graph represents the mean ± s.e. of two independent replicates.

D HCT116 FUT9 knockdown or control cells were injected subcutaneously into the right flank of NOD/SCID mice and monitored for tumor formation. Each tumor was measured using calipers, and the mean volume for the FUT9 knockdown and control tumors was graphed (first panel). The graph represents two independent experiments with a minimum of 11 mice analyzed per experimental condition. Mean tumor volumes ± s.d. are shown. Upon experiment termination, tumors were extracted, weighed, and the mean tumor weights ± s.d. are shown in the second panel.

E A schematic showing the abundance of FUT9-positive cells over the course of colon cancer development.

Data information: *$P$ < 0.05, Student's $t$-test.

**Figure 4.**

N-acetylgalactosamine to a serine or threonine residue of a polypeptide (Freire & Osinaga, 2003). In fact, these N-acetylgalactosaminyl transferases were also found to be highly expressed in different tumors (Milde-Langosch, 2015; Nogimori et al, 2016). Similarly, we found the beta-1,3-N-acetylglucosaminyltransferase B3GNT8 to be upregulated (fold change = 3.46), in FUT9 knockdown cells, consistent with previous reports in colon cancer (Ishida et al, 2005). Overall, these observations are in line with our experimental data and computational predictions that FUT9 inactivation plays a tumorigenic role at later stages, which mostly require invasive and migratory activity.

### FUT9 activity benefits colorectal cancer tumor-initiating cells

Our genomic analysis revealed that, while FUT9 is strongly downregulated at later stages of colon cancer development, it is still present in colon polyps and early adenoma (Appendix Fig S1), indicating that FUT9 activity may be required at the initial stages of tumor initiation. Thus, while FUT9 downregulation benefits the bulk of tumor cells as shown above, its activity may support the subpopulation of cancer stem cells or tumor-initiating cells (TICs) that play a central role in tumor development. To study this hypothesis, HCT116 with FUT9 knockdown and matching control cells were cultured as tumorspheres, which are predominantly formed by TICs (Chan et al, 2016; Liu et al, 2016; Rybicka & Król, 2016; Qureshi-Baig et al, 2017). Consistent with our expectations, FUT9 knockdown reduced expansion of HCT116 cells in tumorspheres, while FUT9 overexpression produced enhanced proliferation of tumorsphere-forming cells (Fig 4A and B). On a molecular level, this was accompanied by the reduced expression of OCT4 transcription factor in FUT9-silenced cells (Appendix Fig S4). Since OCT4 has been shown to support TIC formation (Levings et al, 2009; Chiou et al, 2010), this observation provides a mechanistic explanation for FUT9 effect in supporting TIC activity. These results show that, in contrast to the anti-proliferative effects of FUT9 activity in the bulk of colon cancer cells (Fig 3A–D), FUT9 activity may actually be required for the efficient expansion of TIC populations. This was further confirmed by flow cytometry analysis, showing that FUT9 silencing decreases the expression of a prominent colorectal cancer TIC marker CD44 (Dalerba et al, 2007; Qureshi-Baig et al, 2017) in HCT116 cells (Fig 4C).

### Testing FUT9 activity in a mouse model

Since TIC cells are essential for tumor initiation, tumor maintenance, and tumor growth (Cheng & O'Neill, 2009; Ricci-Vitiani et al, 2009; Grinshtein et al, 2011; Beck & Blanpain, 2013; Bansal et al, 2016; Zhang et al, 2016), increased TIC activity is expected to accelerate tumor growth in vivo (Grinshtein et al, 2011; Beck & Blanpain, 2013; Bansal et al, 2016). To test the effect of FUT9 on this process, we generated a xenograft model of colorectal cancer in immune-deficient NOD/SCID gamma mice. HCT116 cells with silenced FUT9 expression or control cells transduced with non-targeting shRNA were injected subcutaneously in equal numbers into the flank of the immunodeficient mice, and the growth of the resulting tumors was monitored. In agreement with its inhibitory effect in tumorspheres, FUT9 silencing also significantly reduced growth of xenograft tumors (Fig 4D). This may reflect the dual functionality of FUT9 where it supports tumor development by enhancing TIC activity

(Fig 4E), while inhibiting the expansion of the bulk of tumor cells (Fig 3).

## Discussion

We present a novel approach for identifying metabolic tumor suppressors that leads to the discovery of the complex, multi-faceted role of FUT9 in colon cancer. On the methodological side, we show here that a metabolic modeling MTA analysis can successfully identify metabolic genes that play a causal role in cancer initiation and progression from an initial list of genes that are formed via a standard genome-wide analysis. Such an analysis may be thus performed to further identify causal metabolic cancer genes given any list of candidate cancer drivers emerging from a genomic analysis, in other cancer types.

The role of FUT9 in colorectal cancer appears to be rather complex. Our results indicate that FUT9 activity promotes the expansion of TICs, while its downregulation supports expansion and aggressiveness of bulk of tumor cells. TICs represent a higher proportion of the overall cell population in a tumor at earlier stages of tumor development. At later stages however, TICs are gradually outgrown by the rest of the tumor cells (Fig 4E), but they are still required for efficient tumor growth and maintenance (Cheng & O'Neill, 2009; Ricci-Vitiani et al, 2009; Grinshtein et al, 2011; Beck & Blanpain, 2013; Bansal et al, 2016; Zhang et al, 2016). Since our experimental data suggest that FUT9 provides an advantage for TIC populations, while its reduced activity benefits other tumor cells, its relative abundance should be expected to gradually drop with tumor progression, mirroring a decrease in the proportional representation of TICs. Notably, in accordance with that, we found that FUT9 expression is maintained in earlier tumors: colorectal polyps and colorectal adenoma at the levels observed in healthy colon tissue (studied in paired, matched samples; Appendix Fig S1), while FUT9 levels progressively decrease from the M0 to M1 stages (Appendix Fig S5). Reduced FUT9 expression at the M1 metastatic stage also matches our observations, suggesting that FUT9 downregulation enhances migration of colorectal cancer cells. This further supports a notion that as tumors develop, FUT9 activity is switched off in the bulk of tumor cells to enhance their aggressiveness, which should negatively affect patient survival. In agreement, our computational analysis showed a positive correlation between FUT9 expression and survival of colorectal cancer patients.

This study is focused on the identification of tumor suppressor genes, as simulating a gene's knockdown in the metabolic model is very well defined, while simulating the over-expression of genes is more complex and challenging. Thus, developing an MTA approach to identify causal metabolic oncogenes whose overexpression is transforming the metabolic state remains an open challenge. Cancer evolution usually involves a sequence of genetic and environmental events; indeed, while our computational analysis points to the central role that FUT9 plays in generating a tumorigenic metabolic state in colon cancer, we find that its role depends on the overall genomic context, such as the cell types in which it occurs and the staging of the tumors. In agreement, our experimental data reveal that, while FUT9 activity enhances OCT4 expression and is essential for the formation of tumor-initiating cells, it also shows that FUT9

downregulation enhances the invasive behavior of bulk colon cancer cells, which hence contributes at later stages following tumor initiation. Hence, our results should be viewed bearing this reservation in mind.

Overall, our findings support a dual role for FUT9 in colorectal cancer. They suggest that it may act in this malignancy in a manner similar to the reported actions of the EphB2 receptor, a known hallmark of colorectal cancer TICs (Merlos-Suárez *et al*, 2011) that is also downregulated to allow colorectal cancer tumor progression (Cortina *et al*, 2007). Our description of this complex action of FUT9 identifies an entirely new player in colorectal cancer and adds another intriguing member to the rather short list of metabolic genes that have been shown to play a critical role in tumor biology.

# Materials and Methods

### Kaplan–Meyer survival analysis of potential tumor suppressors

For each gene found to be significantly lowly expressed and deleted through gene expression and copy number data, we applied Kaplan–Meyer survival analysis to examine the association of its downregulation with poor patient survival. We use TCGA COAD survival and gene expression data and separate the expression of each gene to "high" and "low" bins by its median level. We calculate the $\Delta AUC$ resulting from the two Kaplan–Meyer curves and select only genes with $\Delta AUC < 0$ indicating that their low expression is associated with poor survival.

### A constraint-based model of metabolism

A metabolic network consisting of $m$ metabolites and $n$ reactions can be represented by a *stoichiometric matrix S*, where the entry $S_{ij}$ represents the stoichiometric coefficient of metabolite $i$ in reaction $j$. A constraint-based model (CBM) imposes mass balance, directionality, and flux capacity constraints on the space of possible fluxes in the metabolic network's reactions through a set of linear equations

$$S \cdot v = 0 \tag{1}$$

$$v_{\min} \leq v \leq v_{\max} \tag{2}$$

Where $v$ is the flux vector for all reactions in the model (i.e., the *flux distribution*). The exchange of metabolites with the environment is represented as a set of *exchange (transport) reactions*, enabling a pre-defined set of metabolites to be either taken up or secreted from the growth media. The steady-state assumption represented in Equation (1) constrains the production rate of each metabolite to be equal to its consumption rate. Enzymatic directionality and flux capacity constraints define lower and upper bounds on the fluxes and are embedded in Equation (2). In the following, flux vectors satisfying these conditions will be referred to as feasible steady-state flux distributions. Gene knockdowns are simulated by constraining the flux through the corresponding metabolic reaction to zero. Similarly, environmental perturbations are simulated by constraining the flux through the associated exchange reaction to zero.

For each of the dataset analyzed here, we simulated the same media that was used in the experiment (DMEM). For modeling human metabolism, we have used Recon1 (Duarte *et al*, 2007).

### Metabolic transformation algorithm (MTA)

MTA receives as input the gene expression measurement of two distinct metabolic states, termed source and target states. Next the algorithm executes the following steps: (i) determine the flux distribution that corresponds to the source state using integration Metabolic Analysis Tool (iMAT); (ii) identify the set of genes whose expression has significantly elevated or reduced between the source and targets states, and the set of genes whose expression remained relatively constant between the states. Next, the algorithm searches for perturbations that can alter the fluxes of the changed reactions in the observed direction, while keeping the fluxes of the unchanged reaction as close as possible to their predicted source state. Finally, MTA outputs a ranked list of candidate perturbations according to their ability to transform from the source to the target metabolic state.

### The transformation score

Relying on the optimization value obtained by MTA to rank the transformations induced by different perturbations is sub-optimal, since the integer-based scoring of the changed reactions is coarse-grained and does not distinguish between solutions achieving large flux alterations and those obtaining flux changes barely crossing the ε threshold. Therefore, we chose to quantify the success of a transformation by a scoring function based on the resulting flux distributions rather than on the optimization objective values themselves. First, we denote the resulting flux distribution obtained in a given MIQP solution (for a given reaction knockout) as $v^{res}$. Second, reactions found in $R_F$ and $R_B$ are classified into two groups $R_{success}$ and $R_{unsuccess}$, denoting whether they achieved a change in flux rate in the required direction (forward or backward) or not. The following scoring function is then used to assess the global change achieved by the employed perturbation:

$$\frac{\sum\limits_{i \in R_{success}} abs\left[\left(v_i^{ref} - v_i^{res}\right)\right] - \sum\limits_{i \in R_{unsuccess}} abs\left[\left(v_i^{ref} - v_i^{res}\right)\right]}{\sum\limits_{i \in R_S} abs\left(v_i^{ref} - v_i^{res}\right)} \tag{3}$$

The numerator of this function is the sum over the absolute change in flux rate for all reactions in $R_{success}$, minus a similar sum for reactions in $R_{unsuccess}$. The denominator is then the corresponding sum over reactions in $R_S$ (the reactions which should stay untransformed). Following, perturbations achieving the highest scores under this definition are the ones most likely to perform a successful transformation by both maximizing the change in flux rate for significantly changed reactions, and minimizing the corresponding change in flux of unchanged reactions. Using an alternative scoring function based on the Euclidean distance instead of absolute values yielded similar results.

While we believe that the TS score (Equation (3)) is the right one to pursue from a biological point of view, optimizing it directly is a very difficult mathematical task. To accomplish that one would need to develop a novel optimization algorithm for

solving a mixed *non-linear* programming problem, whose objective function is non-smooth and non-differentiable, requiring non-smooth optimization tools. Attempting such a solution directly would greatly complicate the problem as one would need to add many variables and constraints. Furthermore, the specific form of this ratio is actually dependent on the solution itself (as it evaluates $R_{success}$ and $R_{unsuccess}$ separately) making the entire task infeasible. In light of these evident difficulties, we have chosen to take a two-step approach in this study that is sub-optimal but yet tractable. While the wild-type solution always achieves maximal values in terms of the original proxy objective function used in step 3 (by definition), it does not necessarily achieve high transformation scores (step 4). This is because the wild-type solution is the least constrained, and hence, most of the solutions found in step 3 can be satisfied by achieving only a minimal epsilon change. Those are obviously non-optimal from a biological standpoint as they do not really come close to the desired objective, and hence, their TS score (in step 4) is sub-optimal in many of the cases, correctly ruling them out as biologically viable solutions. MTA analysis is established upon learning the regulatory effects of the knockdown of metabolic genes via the direct stoichiometric flux coupling of the reactions they encode to other reactions in the human metabolic network (which are inherently embedded in the reactions stoichiometric matrix it includes).

### Aggregated oncogenic transformation scores (OTS)

MTA scores each reaction according to the extent of which its knockout is predicted to cause the observed transformation from normal to cancer. For each reaction $i$ ($RXN_i$), we define the aggregated OTS score by:

$$OTS(RXN_i) = \sum_{j \in matched\ paires} I_{ij} \times \left(1 - P(I_{ij} = 1)\right)$$

Where $I_{ij}$ is one when reaction $i$ was scores higher than random (MTA score when no perturbation is simulated) and zero otherwise. $P(I_{ij} = 1)$ is a reaction's probability to be scored higher than random in matched pair $j$ (which is the number of perturbation that are scored higher then no perturbation in pair $j$). Thus, paired samples in which fewer reactions received a significant score are more heavily weighted.

### Reaction-to-gene mapping of OTS

OTS is assigned to each reaction in the metabolic model. Each metabolic gene is assigned the highest score assigned to one of its associated reactions, using the reaction-to-gene mapping defined by the Recon1 metabolic model.

### Colon polyp and colon tumor gene expression normalization

To apply MTA from polyp to tumor, we applied quantile normalization to the 1,496 metabolic genes present in Recon1 metabolic model. We used 27 colon samples from TCGA that were used for the paired-MTA analysis and 32 colon adenoma sample when the reference distribution is the mean expression of these 1,496 metabolic genes across all 272 colon tumors in TCGA.

### Utilizing MOMA and GIMME algorithms to predict the pathway-level effect of FUT9 inactivation in late-stage colon cancer

To investigate FUT9 role in tumorigenesis in the metabolic model, we set to discover which metabolic flux alterations are induced by the loss of FUT9 in late-stage colon cancer. To this end, we utilized the GIMME algorithm to simulate metabolic flux of stage 3 colon tumors. To evaluate FUT9 effect on metabolic fluxes at that stage, we then utilize the MOMA algorithm and sample 100 flux distributions with and without FUT9 knockdown. For each reaction, we compare the MOMA sampled flux distributions with and without FUT9 KD using one-sided Wilcoxon rank-sum test. We define the set of reactions that are increased following FUT9 knockdown as reactions whose sampled flux is increased when FUT9 knockdown is simulated vs. WT (Wilcoxon rank-sum *P*-value < 0.05) and the set of reactions that are decreased following FUT9 knockdown as reactions whose sampled flux is decreased when FUT9 knockdown is simulated vs. WT (Wilcoxon rank-sum *P*-value < 0.05).

### Utilizing the MOMA algorithm to evaluate the effect of FUT9 knockdown and over-expression on biomass production, glucose consumption, lactate production, and oxygen consumption

To predict the effect of FUT9 levels on biomass production, glucose consumption, lactate production, and oxygen consumption, we utilized the GIMME algorithm to simulate metabolic flux of (i) colon adenoma state using the 32 adenoma samples from Sabates-Bellver *et al* (2007) (ii) Colon cancer state using 268 cancerous samples from the TCGA. For each of the adenoma and cancer predicted flux distributions, we sampled 100 flux distributions for FUT9 KD and another 100 for FUT9 OE (defined by setting the lower bound of FUT9-associated reactions to 80% of their maximum), using MOMA algorithm, aiming to minimize the metabolic adjustments after FUT9 perturbations, from the initial adenoma or cancerous metabolic state. In both cases, we set the lower bound of the biomass reaction to be at least 80% of its optimal rate to simulate proliferating cells and restrict variability in the resulting fluxes.

### Utilizing MOMA algorithm to predict stage-specific context in which the loss of FUT9 is tumorigenic

To predict the context in which the loss of FUT9 drives the oncogenic transformation, we used colorectal cancer gene expression measurements from the TCGA database. For each sample, we predict a flux distribution using the GIMME algorithm (Becker & Palsson, 2008) (the mean flux distribution over 100 sample points was used) and the metabolic model in which FUT9 is knocked down. We then predict a flux distribution typical for stage 4 samples (using the GIMME algorithm (Becker & Palsson, 2008), and genes are considered downregulated with FDR corrected *P*-value < 0.05, compared to all other stages). Then, we compute the MOMA score obtained when aiming to minimize the metabolic adjustment from each sample to the metabolic state predicted for stage 4 samples. Finally, we compare the MOMA score distributions obtained for samples in each of the stages (1–4), describing for each such sample the extent to which the KO of FUT9 is predicted to bring the metabolic flux distribution closer to that of stage 4. A similar analysis

  

was repeated when using iMAT instead of GIMME to predict flux distributions, yielding similar results (Appendix Fig S6).

## Cell lines and transfections

HCT116 and DLD1 colon cancer cell lines were selected based on expression data for FUT9. Both cell lines were cultured in McCoy's 5A medium supplemented with (Fisher Scientific, SH3020001) 10% (v/v) FBS (Life Technologies, 12483020), 100 units/ml penicillin-streptomycin solution (Thermo Scientific, SV30010) at 37°C with 5% $CO_2$. HEK293T cells were used to generate lentivirus and cultured in DMEM (Fisher Scientific, SH3024301) containing 10% (v/v) FBS and 100 units/ml penicillin-streptomycin at 37°C with 5% $CO_2$. Cells were passaged using 0.25% trypsin–EDTA at 70% confluency.

Transfections were done using X-tremeGENE 9 (Roche, 6365809001) as per the manufacturer's instructions. Lentivirus was generated by transfecting HEK293T cells cultured in 100-mm dishes with psPAX2, pMD2.6, and pLKO.1-*sh*RNA or pLX304 expression plasmids. Media was replaced after 24 h with DMEM containing 2% (w/v) bovine serum albumin (BSA) (Fisher Scientific, BP9703100) and lentivirus was harvested after 24 and 48 h and pooled.

To generate the FUT9 knockdown cells, HCT116 and DLD1 cells were transduced with lentivirus containing *sh*RNA sequences specific to FUT9. Two *sh*RNA sequences for FUT9 were used, which were transduced separately or, in subsequent experiments, pooled and transduced together. An *sh*RNA sequence specific to RFP (Sigma) was used as a non-targeting control. For each transduction, 0.5 ml of each *sh*RNA lentivirus was added to $2 \times 10^5$ cells in a 35 mm dish in a final volume of 3 ml with 8 μg/ml of polybrene (Sigma, 107689). Twenty-four hours after transduction, the media was removed and replaced with media containing 2 μg/ml puromycin (Fisher Scientific, BP2956100) for selection. Cells were selected for a minimum of 48 h before use in experiments. Knockdown cells were passaged a maximum of five times. The FUT9-overexpressing cells were generated by transducing HCT116 cells with lentivirus containing pLX304-FUT9 (DNA SU, HsCD00444887) using the same transduction method as above. After transduction, cells were selected using 4 μg/ml of blasticidin (VWR, 89149-988) for 14 days. Cells were maintained with 1 μg/ml of blasticidin.

## Quantitative real-time PCR (RT–qPCR) analysis

RNA was isolated from cell pellets using RNeasy mini kit (Qiagen, 74104) according to the manufacturer's instructions including DNase treatment (Qiagen, 79254). RNA quantification was performed using a NanoDrop 2000c spectrophotometer (Thermo Scientific) and RNA integrity was verified spectrophotometrically by A260/A280 ratios between 1.8 to 2.0 and A260/A230 ratios greater than 1.7. Equal quantities of RNA were used to generate cDNA using the $RT^2$ First Strand Kit (Qiagen, 330401) according to the manufacturer's instructions.

FUT9 expression levels were evaluated using TaqMan real-time PCR gene expression assay (Life Technologies, 4369016 and 4331182, assay ID: Hs00276003_m1). The fold change in gene expression was analyzed using the $\Delta\Delta C_T$ method. Human Glycosylation-related gene expression was evaluated using $RT^2$ Profiler human glycosylation PCR array (Qiagen, 330231 PAHS-046ZA)

according to the manufacturer's instructions. Data analysis was performed using the $\Delta\Delta C_T$ method as described in the manufacturer's web portal (SABiosciences).

## Cell viability assay

Equal numbers of Fut9 knockdown and control cells were seeded in 96 well plates ($5 \times 10^3$ cells per well). After 72 h, the abundance of viable cells was analyzed using Resazurin (Fisher Scientific, AR002). Resazurin was added to each well at a concentration of 10% (v/v) and the plates were incubated at 37°C and read using SpectraMax M5 microplate reader (VWR) after 1, 2, 3, and 4 h. An increased number of viable cells reflect increased cell expansion.

## Growth on soft agar

The ability of FUT9 knockdown and control cells to grow in low-anchorage conditions was determined by seeding cells in a soft agar medium. Cells were trypsinized and $2.5 \times 10^4$ cells suspended in 0.35% agar-media supplemented with 10% (v/v) FBS and 4% (v/v) minimum essential medium vitamin solution (Life Technologies, 11120052) and layered on a 0.6% agar-media bottom layer in 6 well plates. Cells were allowed to grow for 28 days and colonies were imaged using an EVOS FL Cell Imaging System microscope at 40× magnification (Life Technologies) and the density of colonies was quantified using ImageJ software.

## Colony formation assay

The ability of individual cells to form colonies was shown by seeding a low density of cells (50–200 cells per well) in a 24-well culture plate. After 10 days, the colonies were fixed with 100% cold methanol for 10 min and stained using 1% crystal violet. The numbers of visible colonies were counted.

## Wound-healing assay

Cells were cultured in 6-well plates and allowed to grow to a confluent monolayer. A scratch was made in each well by scraping with 100 μl pipette tip across the cell monolayer (time point zero of the experiment). Wells were rinsed with PBS three times to remove floating cells. The same areas of each scratch (2 per scratch) were imaged at the time of scratch (0 h), 24, 48, and 72 h using an EVOS FL Cell Imaging System microscope at 100× magnification. The width of scratch in each image was measured using PowerPoint software.

## Tumorsphere culture and tumorsphere-forming cell counts

For tumorsphere culture, $2 \times 10^3$ cells from monolayer cultures were seeded into 96-well Ultra-Low attachment plates (Corning, 07-200-603) in complete MammoCult medium (Stemcell Technologies, 05620), prepared according to the manufacturer's instruction. Cells were cultured for 7 days, tumorspheres in each well were imaged with an EVOS FL Cell Imaging System microscope. Tumorspheres were then collected, dissociated, and cells were counted using a hemocytometer. For each replicate in this experiment, tumorspheres

from 24 independent wells were collected into a 15 ml tube and centrifuged at $300 \times g$ for 5 min. Collected tumorspheres were dissociated into a single cell suspension in 500 μl of pre-warmed trypsin–EDTA. Cells were washed with tumorsphere culture medium containing 2% FBS and resuspended in serum-free tumorsphere culture medium for cell counting.

### Xenograft models

All animal experimental procedures were reviewed and approved by the University of Saskatchewan Animal Research Ethics Board. Mice used in the present study were from our established colony of NOD SCID gamma mice at the Laboratory Animal Services Unit (LASU), University of Saskatchewan. Mice were maintained at the LASU during the course of the experiments. Control *sh*RFP and *sh*FUT9 knockdown HCT116 cells were trypsinized and resuspended in ice-cold PBS. Cells were mixed 1:1 with Matrigel (Corning, CB-40234) and $3 \times 10^6$ cells in a total volume of 100 μl and injected subcutaneously into the left flank of 6- to 8-week-old immunodeficient NOD/SCID gamma mice. At least five mice that developed tumors were used in our analysis for each experimental condition in each biological replicate. One of the mice in the control group was excluded from the analysis of the last two time points due to lethality. Tumors were measured every 3–4 days using a digital caliper, and the tumor volume was calculated using the tumor ellipsoid formula $A/2*B^2$ where A and B represent the long and the short diameter of the tumor, respectively. Upon experiment termination, tumors were extracted, fixed in 10% formalin, and weighed.

### FACS analysis

Cells were harvested and washed 3 times with ice-cold PBS containing 0.25% FBS. Cells were incubated with FITC-conjugated mouse-anti-human CD44 antibody (BD, 555478) or FITC-conjugated mouse IgG2b antibody (BD, 555742) for 30 min at 4°C in the dark. Cells were then washed thrice with PBS, run through a Beckman Coulter CytoFLEX flow cytometer at 488 nm, and analyzed using CytExpert V1.2 software.

### Data availability

The MTA codes and the GSMM reconstruction are available as Code EV1. The codes and data structures for running MTA from healthy to cancer and from healthy to adenoma are available as Codes EV2 and EV3, respectively. The codes and data structures for generating genomic properties of FUT9 figures are available as Code EV4.

Expanded View for this article is available online.

### Acknowledgement
The authors thank Erez Persi, Welles Robinson, Allon Wagner, and Yonatan Saadon for their comments on the manuscript and helpful discussion.

### Author contributions
NA and ER conceived and designed the research. FJV and AF designed the experimental procedure. NA performed the computational analysis and statistical computations. CEC, BMT, EJM, FSV, SP, TF, and KKB performed the experiments. KY and NG helped with the computational analysis. NA, AF, FJV, and ER wrote the paper.

### Conflict of interest
The authors declare that they have no conflict of interest.

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
