## [Review Process File · Molecular Systems Biology]

An integrated computational and experimental study uncovers FUT9 as a metabolic driver of colorectal cancer

Noam Auslander, Chelsea E. Cunningham, Behzad M. Toosi, Emily J. McEwen, Keren Yizhak, Frederick S. Vizeacoumar, Sreejit Parameswaran, Nir Gonen, Tanya Freywald, Kalpana Kalyanasundaram Bhanumathy, Andrew Freywald, Franco J. Vizeacoumar, Eytan Rupp

Review timeline:

Submission date:	8 May 2017
Editorial Decision:	14 June 2017
Revision received:	4 September 2017
Editorial Decision:	2 October 2017
Revision received:	5 October 2017
Accepted:	7 October 2017

Editor: Maria Polychronidou

Transaction Report:

1st Editorial Decision

14 June 2017

Thank you again for submitting your work to Molecular Systems Biology. We have now heard back from the three referees who agreed to evaluate your study. As you will see below, the reviewers acknowledge that the study seems interesting. However, they raise a series of concerns that we would ask you to address in a major revision of the manuscript.

Without repeating all the points listed below, some of the more fundamental issues are the following:

- Further experimental analyses are required to convincingly support the role of FUT9 as a cancer progression driver and provide some level of mechanistic insight.
- The proposed dual role of FUT9 in the different stages of tumor progression needs to be better supported.
- The essential role of the GSMM analyses for identifying FUT9 as a metabolic driver needs to be better supported/explained.

Of course, all other issues raised by the referees would need to be convincingly addressed.

REVIEWER REPORTS

Reviewer #1:

In the manuscript 'An integrated computational and experimental study uncovers FUT9 as a metabolic driver of colorectal cancer', Auslander et al report the computational identification of FUT9 as potential tumor suppressor whose loss might be causal for colorectal cancer progression. They present preliminary data that demonstrates the stage-dependent, effect of FUT9 loss on colorectal cancer cells and tumors. While I agree that computational modeling of GSMMs integrated with multi-omics datasets is a promising strategy to identify causal mechanisms in inducing metabolic abnormalities in different stages of cancer, this paper's strength the functional characterization of a single gene and not the systems biology. Thus, the scope of study may be more suited for a cancer biology journal. Also from a cancer biology standpoint, more experimental evidence is necessary to support the role of FUT9 in driving cancer progression. I also have some major concerns on the technical details and final outcome of the computational analysis.

1. I'm not convinced of the necessity of applying the GSMM based analysis in identifying FUT9 as the critical metabolic driver. According to Table 1, FUT9 also has nearly the most significant differential expression between normal and cancer so it will also appear as a top hit if only differential expression analysis was used. What is the difference between simulated (and, if possible, experimentally determined) effect on metabolic fluxes of FUT9 knockout and that of deleting other genes with lowest differential expression p-values (e.g. ACADS)?
2. There is some inconsistency between the computational analysis and experimental validation. The authors computed OTS scores for the transition from healthy tissue to adenoma and that from adenoma to cancer, which assess the tendency of the gene deletion to drive the corresponding malignant transformation. It appears that loss of FUT9 might be an important driver not only in adenoma to cancer transformation but also in the earlier healthy tissue to adenoma transformation, according to the high OTS score for FUT9 in the healthy to adenoma transformation. However, in the following experimental results, loss of FUT9 impairs the function of tumorsphere cells, which conflicts with the computational analysis?
3. The authors showed that deletion of FUT9 resulted in enhanced cell proliferation and migration in HCT-116 cells. Thus, it will be helpful if the authors can show if these effects could be correctly predicted by the computational model (e.g. what are the effects of FUT9 deletion on biomass synthesis flux and lactate production flux?)
4. FUT9 deletion is predicted to increase fluxes in pathways such as pentose phosphate pathway and folate metabolism and decrease valine, leucine and isoleucine metabolism, which could be the mechanism for it to benefit cancer cell survival. However, there is no experimental evidence that FUT9 deletion could indeed trigger such a change in cellular metabolism, thus the actual mechanism by which FUT9 knockout impairs cancer cell function remains unclear.
5. FUT9 was found to have dual role in different stages of colorectal cancer progression. I think this is very interesting, but its beneficial role in the earlier stage was less well characterized in this study. What are the effects of FUT9 knockout and over-expression on cellular metabolism during that stage? Although an in-depth metabolomics analysis might be beyond the scope of this study, this could be easily predicted by GSMM-based methods such as MOMA.
6. The method the authors used in computing the OTS scores needs to be explained in more detail. First, it is unclear to me how the MTA scores for each reaction are computed. Second, I'm also not sure about the rationale of using the fraction of significant hits in the MTA step as a weight in computing the OTS scores, since this will bias towards genes predicted by MTA in matched sample pairs with more significant hits. In other words, genes that tend to appear together with other genes as metabolic drivers in certain matched pairs are more likely to have higher OTS scores. If the OTS scores are simply computed by counting total number of matched sample pairs with the corresponding gene as a significant hit by MTA, will FUT9 still have the highest OTS?

Reviewer #2:

In the manuscript by Auslander et al., the authors first computationally identified the gene FUT9 as a tumor suppressor at the later stage of colon cancer, and then experimentally verified the prediction. The two-step computational procedure is novel: while the first step involves traditional gene enrichment analysis, the second step of identifying genes driving one metabolic state to another using metabolic network and expression data is original. The experimental tests are comprehensive, using different cell cultures and xenografted tumors on mice. The study is a good example of integrating computation and experiment. The approach can be applied to other cancer types. I however have a few comments on some unclear points and suggestions for improving the manuscript.

Major:

The study focused on identifying tumor suppressor genes. What about oncogenes? They are equally important. At least the omission should be discussed.

While the MTA analysis is novel and informative, I see some limitations: it tests the effect of individual gene knockouts. But it's well known that tumor results from multiple mutations. This limitation should be discussed.

Similar to Supplementary Tables 1-3, it would be useful for the reader to see the list of genes with top OTS for each data set and also genes with top aggregate OTS.

GIMME and iMAT are both algorithms for predicting flux distributions based on expression data. Why is one used in some places and the other used in other places?

Why is Fig. 2E a prediction? Is it a tautology? The criterion used to identify FUT9 is the closeness between the metabolic state of advanced tumor and the predicted metabolic state with FUT9 knockout.

The regulatory mechanism regarding FUT9 should be discussed.

Minor:

There are several methods with acronym names that should be briefly explained before pointing to references. These include GIMME, MOMA, CN Q-value, and iMAT.

On page 9, "used the MOMA algorithm" should be "using the MOMA algorithm".

On page 14, in the paragraph of xenograft tumors, "Figure 4C" should be "Figure 4D", and "Figure 4" should be "Figure 4E".

Reviewer #3:

The manuscript by Auslander et al entitled, "An integrated computational and experimental study uncovers FUT9 as a metabolic driver of colorectal cancer" presents a systems investigation into genes associated with development and progression of colorectal cancer.

The manuscript is well-written with systematic experimental validation of computational predictions. Although some statements regarding the strength of association and actual causality are a bit overstated, the study is well-designed and provides a convincing example of a systematic experimental and computational systems biology study.

There are a few areas that would benefit from clarification and revision,

- P6, "top 20% of the predictions", what was the criteria or justification for selection of the top 20% (as opposed to 5% or 50%)?

- P7 (table 1), The "distribution of the resulting OTS" is reported, but there isn't any explicitly described assessment of the distribution of the cutoffs for selection of the OTS scores. Although 2.99 and 2.67 are the highest two values, one could just as easily argue that FH (OTS score 1.2), SLC18A2 (OTS score 0.73), and PANK4 (OTS score 1.2), for example, should also be selected (ITPKA presumably could be excluded due to Q-value), since those values are well above the mean and median scores, with statistical significance. Where there any objective criteria that were applied before calculating the results or was this more a selection out of convenience?

- P10, Figure 2C, the reported p-value 0.1942 does not appear consistent with the displayed survival curve. Perhaps a typo?

- P13, "were seeded at very low densities in a 24 well dish and cultured for 10 days". The methods note that these were 50 to 200 cells per well, was there a negative control that reached some level of confluence at day 10?

- P20, "predict a flux distribution using the GIMME algorithm". Technically speaking, a single solution from an FBA simulation is a point in a solution space, so a distribution would be a sampled set of points in this space. It is not clear from the text if a flux distribution were calculated by repeated runs of GIMME or if a single solution were selected. If the latter were performed, there should be an argument for why alternative solution points that satisfy the GIMME problem would not alter the results (it is clear that MOMA solution will be unique, but different input models to MOMA could potentially lead to different calculated distances).

- While the progression from cell culture -> tumorsphere -> xenograft model testing provides increased levels of confidence in the results, the degree to which xenografts reflect disease in situ is recognized to be limited, so this warrants tempering the conclusions, particularly in light of the supplemental figures (small sample sizes, large variance), which require a degree of rationalization and sheds some degree of uncertainty on the strength of the assertion that FUT9 is a singular driver of malignancy. Indeed, seeing a "gradual change" over time (slight but non-significant decrease at adenoma stage to borderline significant change at M1 from M0), would make one more likely to argue that this may be a "co-driver" or potentially even a passenger gene, as opposed to a singular driver. As an alternative to the "dual role" argument for the role of FUT9, the authors should consider alternative hypothesis that FUT9 may not be the sole driver of the transformation to malignancy.

Minor typos that may warrant another read-through of the manuscript,

- p18, "_ij" likely is meant to read "sij"

- p18, Missing period at the end of the sentence, "For modeling human metabolism ..."

- p3, "dual role in this malignancy: its expression ...", may read more fluidly as, "dual role in this malignancy; its expression ..."

- Capitalization (or lack thereof) of "supplementary information" is not consistent throughout the manuscript. Similarly P-value vs p-value.

1st Revision - authors' response

4 September 2017

Reviewer #1:

In the manuscript 'An integrated computational and experimental study uncovers FUT9 as a metabolic driver of colorectal cancer', Auslander et al report the computational identification of FUT9 as potential tumor suppressor whose loss might be causal for colorectal cancer progression. They present preliminary data that demonstrates the stage-dependent, effect of FUT9 loss on colorectal cancer cells and tumors. While I agree that computational modeling of GSMs integrated with multi-omics datasets is a promising strategy to identify causal mechanisms in inducing metabolic abnormalities in different stages of cancer, this paper's strength the functional characterization of a single gene and not the systems biology. Thus, the scope of study may be more suited for a cancer biology journal. Also from a cancer biology standpoint, more experimental evidence is necessary to support the role of FUT9 in driving cancer progression. I also have some major concerns on the technical details and final outcome of the computational analysis.

Relating to the comment regarding its scope, our paper presents a systems biology genome-wide computational method that starts from the analysis of genome wide transcriptomics data of tumors and matched healthy tissues to identify candidate causal metabolic driver genes. The experimental study then focuses and drills down on one top predicted gene, as is customarily done in many systems biology papers (including those published in high impact journals) given the effort involved in the latter.

1. I'm not convinced of the necessity of applying the GSMM based analysis in identifying FUT9 as the critical metabolic driver. According to Table 1, FUT9 also has nearly the most significant differential expression between normal and cancer so it will also appear as a top hit if only differential expression analysis was used. What is the difference between simulated (and, if possible, experimentally determined) effect on metabolic fluxes of FUT9 knockout and that of deleting other genes with lowest differential expression p-values (e.g. ACADS)?

We thank the reviewer for pointing out this important issue. In a nutshell, if we would have selected the candidate genes solely by differential expression, FUT9 would only be ranked no. **124**, and would have been preceded by 20 other metabolic genes, as listed in Table EV1. However, to make the genomic analysis as comprehensive as possible, we additionally (beyond expression) examined each gene's Copy Number Q-value and its Kaplan-Meier survival delta-AUC to select the top candidate genes in the genomic analysis step (those are listed in Table1). 34 metabolic genes satisfy all genomic selection criteria and are then further evaluated via an MTA analysis in the second step. Among these 34 genes, FUT9 indeed has the second highest differential expression p-value, but many other genes have significantly lower Q values or higher KM AUCs, as evident from Table1 in the manuscript.

Beyond that, it is important to note that the MTA analysis is conceptually different from the genomic analysis. While the latter aims to find genes whose altered genomic state is *associated* with the tumorigenic state, MTA is designed to identify metabolic genes whose KD is predicted to *causally* transform the source (healthy) metabolic state to that of the target (cancerous) one. ACADS and other targets that are highly listed in Table1 received lower overall MTA scores (termed OTS) than FUT9; that is, in difference from FUT9, the KD of these genes does not result in predicted metabolic state that is very close to that observed in tumors. Thus FUT9 was selected and further studied experimentally. These issues are now further clarified in the text on page 7.

2. There is some inconsistency between the computational analysis and experimental validation. The authors computed OTS scores for the transition from healthy tissue to adenoma and that from adenoma to cancer, which assess the tendency of the gene deletion to drive the corresponding malignant transformation. It appears that loss of FUT9 might be an important driver not only in adenoma to cancer transformation but also in the earlier healthy tissue to adenoma transformation, according to the high OTS score for FUT9 in the healthy to adenoma transformation. However, in the following experimental results, loss of FUT9 impairs the function of tumorsphere cells, which conflicts with the computational analysis?

We thank the reviewer for this comment. Indeed, based on MTA analysis FUT9 was predicted as a driver both for the healthy to adenoma transformation and for the adenoma to cancer transformation. However: (1) based on the genomic analysis, we did not observe any decrease in FUT9 expression in colon adenomas vs. healthy tissues (Appendix Figure S1).

We now explicitly explain this issue further in the revised manuscript (page 7), as follows:

“Interestingly though, while MTA highly scores FUT9 for all three transformations, FUT9 is not significantly downregulated at early stage colon adenomas using paired gene expression of healthy/adenoma samples from Sabates-Bellver et al.”

Appendix Figure S1- boxplots showing the expression of FUT9 in adenoma vs. healthy samples.

(2) Following up on the reviewer's comment, we now performed an additional GSMM analysis to evaluate the stage-specific effects of the loss of FUT9, and find that it is much more likely to cause the oncogenic transformation in later stages rather than in early stages (Figures 2E, 2F and 2G)

3. The authors showed that deletion of FUT9 resulted in enhanced cell proliferation and migration in HCT-116 cells. Thus, it will be helpful if the authors can show if these effects could be correctly predicted by the computational model (e.g. what are the effects of FUT9 deletion on biomass synthesis flux and lactate production flux?)

We thank the reviewer for this suggestion. To evaluate these, we simulated the flux distribution of the cancerous state (using GIMME algorithm) and sampled 100 flux distributions using MOMA with FUT9 KD and OE. We compared the biomass production, glucose consumption, lactate production and oxygen consumption rates between the simulated adenoma and cancerous metabolic states after FUT9 KD and OE. We have added this analysis to the revised manuscript and the relevant new text and figure panel now reads:

“We next evaluated the metabolic effects of FUT KD and OE in the colon tumor state. To this end we performed a similar analysis as described above for adenoma, while first inferring the likely metabolic state of colon tumors (Methods). Strikingly, we find that the predicted biomass production in the cancerous state is significantly higher under FUT9 KD than its OE (Wilcoxon rank-sum P-value = 0.0245, Figure 2F), and that lactate production rate is also increased under FUT9 KD (Wilcoxon rank-sum P-value = 0.0859, Figure 2F), opposite to the observed in simulated colon adenoma state. These predictions imply that the loss of FUT9, while hampering the growth of adenomas, is required for the proliferation of colon tumors, while its overexpression significantly reduces proliferation in that state.” (Pages 9-10).

Figure 2(F) Boxplot showing the distribution of biomass production, Glucose consumption, Lactate production and Oxygen consumption in cancer state when FUT9 is knocked-down (KD) and overexpressed (OE).

4. FUT9 deletion is predicted to increase fluxes in pathways such as pentose phosphate pathway and folate metabolism and decrease valine, leucine and isoleucine metabolism, which could be the mechanism for it to benefit cancer cell survival. However, there is no experimental evidence that FUT9 deletion could indeed trigger such a change in cellular metabolism, thus the actual mechanism by which FUT9 knockout impairs cancer cell function remains unclear.

We thank the reviewer for this comment and have now performed additional computational analyses and experiments to assess the role of FUT9 in metabolic pathways and cancer aggressiveness, as follows:

(1) First, we changed the GSMM pathway analysis and performed a stage specific analysis (Using GIMME and MOMA algorithms) to predict the pathway-level effects of the loss of FUT9 in late stage. This analysis point to key enzymes in the TCA cycle whose expression is predicted to increase following the loss of FUT9 in late stage (stage 3) colon cancer. We find that the expression of these genes is indeed elevated from stage 3 to stage 4 colon cancer in TCGA data, and that the loss of FUT9 results in the up-regulation of these genes' expression, as now corroborated experimentally. These data are now included in the revised version of the manuscript and summarized in Fig2D with its corresponding data shown in the Supp. Fig.2.

(2) Second, we now show that FUT9 affects the glycosylation pathway. In particular, we find that the expression of GANAB and GCNT3 is downregulated while GALNT8, GALNT12, GALNT13 and B3GNT8 are upregulated in FUT9 silenced cells, as predicted by the MTA analysis. Since the loss of GANAB and GCNT3 has been shown to increase cancer aggressiveness, and overexpression of GALNT8, GALNT12, GALNT13 and B3GNT8 has been reported in colon cancer, these results show that the changes occurring in the glycosylation pathway (Supp. Fig.3) may account, at least partially, for the function of its KD in supporting colon cancer aggressiveness.

5. FUT9 was found to have dual role in different stages of colorectal cancer progression. I think this is very interesting, but its beneficial role in the earlier stage was less well characterized in this study. What are the effects of FUT9 knockout and over-expression on cellular metabolism during that stage? Although an in-depth metabolomics analysis might be beyond the scope of this study, this could be easily predicted by GSMM-based methods such as MOMA.

We thank the reviewer for this helpful suggestion. Following, we now performed a MOMA metabolic modeling analysis to predict the metabolic states after the KO and OE of FUT9, at the initial colon adenoma state (similar to the analysis performed at the initial cancerous state). We find that the KO of FUT9 results in reduced biomass production and lactate secretion rates compared to its OE in adenoma stage, implying that in the adenoma state FUT9 deletion is likely to reduce proliferation and Warburg effect when comparing the to

over-activation of FUT9, opposite to its predicted effect in cancerous state. We have added these results to the revised manuscript in Figure 2E (page 9):

“To evaluate the effect of FUT knockdown (KD) and overexpression (OE) on biomass production, Glucose consumption, Lactate production and Oxygen consumption in the benign colon adenoma state, we (1) simulated the wild-type metabolic state associated with colon adenoma. This was done by incorporating adenoma gene expression data from Sabates-Bellver et al.³⁸ using the Gene Inactivity Moderated by Metabolism and Expression (GIMME) algorithm. (2) We then sampled 100 flux distributions in the resulting predicted adenoma wild-type state. In each such sample we applied the Minimization Of Metabolic Adjustment (MOMA)⁵⁰ algorithm to predict the metabolic state after FUT9 KD and OE in adenoma, summing up the results overall 100 samples (Methods). We find that the biomass production predicted is significantly higher under FUT OE than its KD, as well as Lactate secretion rate (Wilcoxon rank-sum P-value = 0.0081 and 0.0173, respectively, Figure 2E), while oxygen consumption rate is significantly higher under FUT9 KD (Wilcoxon rank-sum P-value = 6.79e-8, Figure 2E). These predictions imply that FUT9 activity is required for supporting cancer proliferation in the adenoma state, which are consistent with the genomic findings we reported above that while FUT9 expression is strongly downregulated in colon cancer is not significantly downregulated at early stage colon adenomas.”

6. The method the authors used in computing the OTS scores needs to be explained in more detail. First, it is unclear to me how the MTA scores for each reaction are computed. Second, I'm also not sure about the rationale of using the fraction of significant hits in the MTA step as a weight in computing the OTS scores, since this will bias towards genes predicted by MTA in matched sample pairs with more significant hits. In other words, genes that tend to appear together with other genes as metabolic drivers in certain matched pairs are more likely to have higher OTS scores. If the OTS scores are simply computed by counting total number of matched sample pairs with the corresponding gene as a significant hit by MTA, will FUT9 still have the highest OTS?

Thanks for these comments. We apologize for not being sufficiently clear about this in the previous version. As to your first comment, the MTA transformation scores are calculated in exactly the same way as described in Yizhak et al.(1); We have now added this description to the Methods section to make the revised manuscript self-contained. It reads as follows (page 21):

“The Transformation Score

Relying on the optimization value obtained by MTA to rank the transformations induced by different perturbations is suboptimal, since the integer-based scoring of the changed reactions is coarse-grained and does not distinguish between solutions achieving large flux alterations and those obtaining flux changes barely crossing the ϵ threshold. Therefore, we chose to quantify the success of a transformation by a scoring function based on the resulting flux distributions rather than on the optimization objective values themselves. First, we denote the resulting flux distribution obtained in a given MIQP solution (for a given reaction knock-out) as v^{res} . Second, reactions found in R_F and R_B are classified into two groups $R_{success}$ and $R_{unsuccess}$, denoting whether they achieved a change in flux rate in the required direction (forward or backward) or not. The following scoring function is then used to assess the global change achieved by the employed perturbation:

$$\frac{\sum_{i \in R_{success}} abs[(v_i^{ref} - v_i^{res})] - \sum_{i \in R_{unsuccess}} abs[(v_i^{ref} - v_i^{res})]}{\sum_{i \in R_S} abs(v_i^{ref} - v_i^{res})} \quad (10)$$

The numerator of this function is the sum over the absolute change in flux rate for all reactions in $R_{success}$, minus a similar sum for reactions in $R_{unsuccess}$. The denominator is then the corresponding sum over reactions in R_S (the reactions which should stay untransformed). Following, perturbations achieving the highest scores under this definition are the ones most likely to perform a successful transformation by both maximizing the change in flux rate for significantly changed reactions, and minimizing the corresponding change in flux of unchanged reactions. Using an alternative scoring function based on the Euclidean distance instead of absolute values yielded similar results.

While we believe that the TS score (Equation (10)) is the right one to pursue from a biological point of view, optimizing it directly is a very difficult mathematical task. To

accomplish that one would need to develop a novel optimization algorithm for solving a mixed *non-linear* programming problem, whose objective function is non-smooth and non-differentiable, requiring non-smooth optimization tools. Attempting such a solution directly would greatly complicate the problem as one would need to add many variables and constraints. Furthermore, the specific form of this ratio is actually dependent on the solution itself (as it evaluates $R_{success}$ and $R_{unsuccess}$ separately) making the entire task infeasible. In light of these evident difficulties we have chosen to take a two-step approach in this study that is sub-optimal but yet tractable. While the wild-type solution always achieves maximal values in terms of the original proxy objective function used in step 3 (by definition), it does not necessarily achieve high transformation scores (step 4). This is because the wild type solution is the least constrained, and hence most of the solutions found in step 3 can be satisfied by achieving only a minimal epsilon change; Those are obviously non-optimal from a biological standpoint as they do not really come close to the desired objective, and hence their TS score (in step 4) is sub-optimal in many of the cases, correctly ruling them out as biologically viable solutions.”

As to your second comment, indeed there was typo in the text of the methods section describing the OTS score, for which we sincerely apologize – thanks much (and indeed, this is was not the way the OTS was actually calculated). The OTS gives **higher** weight to pairs in which **fewer** reactions were significantly scored (as in such cases this event is more unique). However, even if we would not have used any weights, the ranking of the targets remains quite close to that shown in Table1 (and FUT9 is still the top predicted target, along with 2 others).

The Methods section has been corrected in the revised manuscript accordingly (page 22):

“**Aggregated oncogenic transformation scores (OTS)**. MTA scores each reaction according to the extent of which its knockout is predicted to cause the observed transformation from normal to cancer. For each reaction i (RXN_i) we define the aggregated OTS score by:

$$OTS(RXN_i) = \sum_{j \in \text{matched paires}} I_{ij} \times (1 - P(I_{ij} = 1))$$

Where I_{ij} is one when reaction i was scores higher than random (MTA score when no perturbation is simulated) and zero otherwise. $P(I_{ij} = 1)$ is a reaction’s probability to be scored higher than random in matched pair j (which is the number of perturbations that are scored higher then no perturbation in pair j). Thus, paired samples in which less reaction received a significant score are more heavily weighted. “

Reviewer #2:

In the manuscript by Auslander at al., the authors first computationally identified the gene FUT9 as a tumor suppressor at the later stage of colon cancer, and then experimentally verified the prediction. The two-step computational procedure is novel: while the first step involves traditional gene enrichment analysis, the second step of identifying genes driving one metabolic state to another using metabolic network and expression data is original. The experimental tests are comprehensive, using different cell cultures and xenografted tumors on mice. The study is a good example of integrating computation and experiment. The approach can be applied to other cancer types. I however have a few comments on some unclear points and suggestions for improving the manuscript.

We thank the reviewer for his positive and constructive review of our work – much appreciated.

Major:

The study focused on identifying tumor suppressor genes. What about oncogenes? They are equally important. At least the omission should be discussed.

We agree. However, to use MTA to identify oncogenes one would have to simulate the over-expression of genes, which is more complex and less well-defined than simulating gene deletion in GSMMs. We now discuss this challenge in brief in the discussion section:

“This study is focused on the identification of tumor suppressor genes, as simulating a gene’s knockdown in the metabolic model is very well defined, while simulating the over-expression of genes is more complex and challenging. Thus, developing an MTA approach to identify causal metabolic oncogenes whose overexpression is transforming the metabolic state remains an open challenge” (Page 19).

While the MTA analysis is novel and informative, I see some limitations: it tests the effect of individual gene knockouts. But it's well known that tumor results from multiple mutations. This limitation should be discussed.

Agreed. We show that the effects of FUT9 KD are indeed stage dependent (pages 9-10) and following the reviewer’s comment we now note in brief in the discussion section (page 19):

“Cancer evolution usually involves a sequence of genetic and environmental events; indeed, while our analysis points to the central role that FUT9 plays in generating a tumorigenic metabolic state in colon cancer, we find that its role depends on the overall genomic context, such as the cell types in which it occurs and the staging of the tumors. Hence, our results should be viewed bearing this reservation in mind.

Similar to Supplementary Tables 1-3, it would be useful for the reader to see the list of genes with top OTS for each data set and also genes with top aggregate OTS.

Thanks. We added a second list including OTS for all metabolic genes studied in the 4 datasets to Table EV4, which previously has included OTS values only for the 33 metabolic genes predicted via the first genomic step.

GIMME and iMAT are both algorithms for predicting flux distributions based on expression data. Why is one used in some places and the other used in other places?

We apologize for not sufficiently explicating this issue previously. iMAT is the first step of MTA (and is built in the algorithm) and hence it was used for all MTA analysis. We used GIMME for the MOMA analysis, as it requires significantly less run time. To emphasize the robustness of the latter analysis, we have repeated it using iMAT instead of GIMME to predict the flux distributions. The results of this analysis are now reported in the Appendix (Appendix Figure S6). It is now referred to from the Methods section (page 20):

“A similar analysis was repeated when using iMAT instead of GIMME to predict flux distributions, yielding similar results (Appendix Figure S6).”

Why is Fig. 2E a prediction? Is it a tautology? The criterion used to identify FUT9 is the closeness between the metabolic state of advanced tumor and the predicted metabolic state with FUT9 knockout.

The initial, basic analysis, *compared the effects of the KD FUT9 to all other 32 metabolic genes* examined at the second stage and established that, among those, FUT9 KD is the most likely to drive the healthy/adenoma metabolic state to that of colorectal tumor. The second, stage-dependent analysis is different and more refined. *It focuses just on FUT9, and studies the effects of its KD in each of four different stages.* The text has now been modified to make this more explicit (page 10):

“We performed an additional GSMM analysis to study whether FUT9 inactivation at early colorectal cancer stages can induce the metabolic state observed at advanced tumors, or only its inactivation at late stages can induce this transformation. To this end we first inferred the likely metabolic state of advanced colorectal tumors using the GIMME

algorithm⁴⁶, as done above in the adenoma analysis. We then predicted the likely metabolic states after the loss of FUT9 in each of the four different stages of colorectal cancer progression, asking how similar is the metabolic state induced after the loss of FUT9 in each of these stages to the advanced, late cancerous state.”

The regulatory mechanism regarding FUT9 should be discussed.

The totality of regulatory effects of FUT9, like that of most other human metabolic genes, are yet unknown. Given this state of affairs, our analysis is based on studying the regulatory effects that the metabolic genes have via direct stoichiometric flux coupling to other reactions in the human metabolic network (which are inherently embedded in the reactions stoichiometric matrix it includes). We added a note to this effect in the description of MTA in the Methods section, which reads (page 22)

“MTA analysis is established upon learning the regulatory effects of the knockdown of metabolic genes via the direct stoichiometric flux coupling of the reactions they encode to other reactions in the human metabolic network (which are inherently embedded in the reactions stoichiometric matrix it includes).”

Minor:

There are several methods with acronym names that should be briefly explained before pointing to references. These include GIMME, MOMA, CN Q-value, and iMAT.

We thank the reviewer for pointing this out. All acronym names are now explained at their first appearance in the manuscript:
Copy Number (CN) deletion Q-value (P-value that has been adjusted for the False Discovery Rate) (page 8), Gene Inactivity Moderated by Metabolism and Expression (GIMME) and Minimization Of Metabolic Adjustment (MOMA) (page 9) and integration Metabolic Analysis Tool (iMAT) (page 21).

On page 9, "used the MOMA algorithm" should be "using the MOMA algorithm".

On page 14, in the paragraph of xenograft tumors, "Figure 4C" should be "Figure 4D", and "Figure 4" should be "Figure 4E".

We thank the reviewer for pointing these out, both has been corrected in the revised version of the manuscript.

Reviewer #3:

The manuscript by Auslander et al entitled, "An integrated computational and experimental study uncovers FUT9 as a metabolic driver of colorectal cancer" presents a systems investigation into genes associated with development and progression of colorectal cancer.

The manuscript is well-written with systematic experimental validation of computational predictions. Although some statements regarding the strength of association and actual causality are a bit overstated, the study is well-designed and provides a convincing example of a systematic experimental and computational systems biology study.

We thank the reviewer for his positive and constructive review of our work – much appreciated.

There are a few areas that would benefit from clarification and revision,

- P6, "top 20% of the predictions", what was the criteria or justification for selection of the top 20% (as opposed to 5% or 50%)?

Indeed, this choice has been somewhat arbitrary. We now show in the Appendix that these results remain robust for a choice of 5,10 and 15 % of the predictions as well. The main text now reads (page 6):

“(Binomial P-value = 5.8266e-06, and remain robust for other threshold settings, Appendix)”

- P7 (table 1), The "distribution of the resulting OTS" is reported, but there isn't any explicitly described assessment of the distribution of the cutoffs for selection of the OTS scores. Although 2.99 and 2.67 are the highest two values, one could just as easily argue that FH (OTS score 1.2), SLC18A2 (OTS score 0.73), and PANK4 (OTS score 1.2), for example, should also be selected (ITPKA presumably could be excluded due to Q-value), since those values are well above the mean and median scores, with statistical significance. Where there any objective criteria that were applied before calculating the results or was this more a selection out of convenience?

We thank the reviewer for pointing this out. The criteria used are: (1) A gene that is highly scored in the healthy → cancer OTS score. (2) A gene that is highly scored in the adenoma → cancer OTS score. The two genes that best satisfy these two criteria are PTEN and FUT9, and as PTEN is already known and studied, we set to further investigate the FUT9 gene.

To clarify this point the manuscript now reads (page 7):

“As evident, only the knockdown of PTEN and FUT9 is predicted to transform the metabolic state of healthy cells as well as that of adenoma cells to that of colorectal tumors with high OTS scores (Methods).”

Yet, it is certainly possible that other genes that rank high in our analysis may play a causative transformative role in colon cancer and may be worthy of further experimental study in the future. We now added a brief note to this extent to the discussion – thanks (on page 19):

“We focused here on the investigation of FUT9 as it rose as a top predicted target in our analysis; nevertheless, it is highly possible that other targets listed here have a causal role in driving cancer, and should be explored in future studies.”

- P10, Figure 2C, the reported p-value 0.1942 does not appear consistent with the displayed survival curve. Perhaps a typo?

This is not a typo. This relatively high p-value is probably a result of the fact that only 39 samples FUT9 has non-zero expression (that composes the entire red curve), meaning that 85% of the (overall 268) patients has zero FUT9 expression (composing the blue curve, see the histogram of COAD FUT9 expression here). As a result, while the delta-AUC is marked, the p-value is not as significant. We now briefly discuss this:

“The resulting KM log-rank P-value is 0.1942, likely due to the small sample size of patients expressing FUT9 (only ~15% of patients). (Page 7)

Histogram of FUT9 expression in the TCGA dataset

- P13, "were seeded at very low densities in a 24 well dish and cultured for 10 days". The methods note that these were 50 to 200 cells per well, was there a negative control that reached some level of confluence at day 10?

In this experiment, we are counting colonies formed by cells with silenced FUT9 expression and a negative control cells transduced with shRFP. When cells reach confluence, colony counting is not possible. Therefore, our experiments were designed in a manner where none of our cells reached confluence.

- P20, "predict a flux distribution using the GIMME algorithm". Technically speaking, a single solution from an FBA simulation is a point in a solution space, so a distribution would be a sampled set of points in this space. It is not clear from the text if a flux distribution were calculated by repeated runs of GIMME or if a single solution were selected. If the latter were performed, there should be an argument for why alternative solution points that satisfy the GIMME problem would not alter the results (it is clear that MOMA solution will be unique, but different input models to MOMA could potentially lead to different calculated distances).

We apologize for not clarifying this point sufficiently. The flux distribution used for each sample is the mean of 100 flux distributions that were randomly sampled via GIMME algorithm. The text now explicitly states this as follows (page 24):

“For each sample, we predict a flux distribution using the GIMME algorithm (the mean flux distribution over 100 sample points was used) and the metabolic model in which FUT9 is knocked down.”

- While the progression from cell culture -> tumorsphere -> xenograft model testing provides increased levels of confidence in the results, the degree to which xenografts reflect disease in situ is recognized to be limited, so this warrants tempering the conclusions, particularly in light of the supplemental figures (small sample sizes, large variance), which require a degree of rationalization and sheds some degree of uncertainty on the strength of the assertion that FUT9 is a singular driver of malignancy. Indeed, seeing a "gradual change" over time (slight but non-significant decrease at adenoma stage to borderline significant change at M1 from M0), would make one more likely to argue that this may be a "co-driver" or potentially even a passenger gene, as opposed to a singular driver. As an alternative to the "dual role" argument for the role of FUT9, the authors should consider alternative hypothesis that FUT9 may not be the sole driver of the transformation to malignancy.

We thank the reviewer for this important comment. We absolutely agree that FUT9 may not be a singular driver; we only argue that it is an important metabolic driver of colon cancer, but certainly it does not act alone. The importance of FUT9 in tumor development is now further confirmed by our new data showing that FUT9 increases expression the OCT4 transcription factor, which has been previously shown to induce the formation of tumor initiating cells. These data are now shown in Appendix figure S4 of the revised manuscript. Moreover, the role of FUT9 is context and cell dependent, as our analyses clearly show. We now explicitly highlight this point in the discussion section (page 19):

“This study is focused on the identification of tumor suppressor genes, as simulating a gene’s knockdown in the metabolic model is very well defined, while simulating the over-expression of genes is more complex and challenging. Thus, developing an MTA approach to identify causal metabolic oncogenes whose overexpression is transforming the metabolic state remains an open challenge. Cancer evolution usually involves a sequence of genetic and environmental events; indeed, while our computational analysis points to the central role that FUT9 plays in generating a tumorigenic metabolic state in colon cancer, we find that its role depends on the overall genomic context, such as the cell types in which it occurs and the staging of the tumors. In agreement, our experimental data reveal that, while FUT9 activity enhances OCT4 expression, and is essential for the formation of tumor initiating cells, it also show that FUT9 downregulation enhances the invasive behavior of bulk colon cancer cells, which hence contributes at later stages following tumor initiation. Hence, our results should be viewed bearing this reservation in mind.” (Page 19)

Minor typos that may warrant another read-through of the manuscript,

- p18, "_ij" likely is meant to read "sij"
- p18, Missing period at the end of the sentence, "For modeling human metabolism ..."
- p3, "dual role in this malignancy: its expression ...", may read more fluidly as, "dual role in this malignancy; its expression ..."
- Capitalization (or lack thereof) of "supplementary information" is not consistent throughout the manuscript. Similarly P-value vs p-value.

We thank the reviewer for pointing these out, and corrected all in the revised version accordingly.

2nd Editorial Decision

2 October 2017

Thank you again for sending us your revised manuscript. We have now heard back from the two referees who were asked to evaluate your study. As you will see below, the reviewers think that most of the previously raised issues have been satisfactorily addressed and that the study is now suitable for publication.

We have additionally consulted with Reviewer #3 on whether, in their opinion, the information provided in the manuscript is sufficient to allow others to reproduce the computational analyses reported in the study. To this s/he replied: "Following further detailed review of the supplemental material, it is completely reasonable to request that the authors provide at least one (if not both) of the following in order to enable accessible reproducibility of the presented results:

1) A script that implements a pipeline to carry out the computational workflow in order to generate the models and results in the Figures (notably Table 1 and Fig 2) and 2) GSMM models that were constructed and used in the analyses available in SBML format (or similarly accessible format that would provide utility with current systems biology tools and programs)."

In line with this comment, we would ask you to provide both the script implementing the computational workflow for building the models and the GSMMs in SBML format. They should be provided as Computer Code EV1, EV2 etc. as .zip folders. Please include a README.txt file in each of the Computer Code .zip folders providing a short description and any additional details and/or instructions if necessary. These files need to be mentioned in a Data Availability section at the end of the Materials and Methods.

REVIEWER REPORTS

Reviewer #1:

The authors have addressed my technical concerns about the modeling and have done some more experiments. I still think the cancer biology aspects of this paper are not well developed which is the focus of this paper but i'm happy to leave that to editorial discretion as to whether this paper is a sufficient advance in systems biology.

Reviewer #3:

The previously raised concerns have been addressed.

Additional comments from Reviewer #3:

"Following further detailed review of supplemental material, it is completely reasonable to request that the authors provide at least one (if not both) of the following in order to enable accessible reproducibility of the presented results:

1) A script that implements a pipeline to carry out the computational workflow in order to generate the models and results in the Figures (notably Table 1 and Fig 2) and 2) GSMM models that were

constructed and used in the analyses available in SBML format (or similarly accessible format that would provide utility with current systems biology tools and programs)."

2nd Revision - authors' response

5 October 2017

Reviewer #3: Following further detailed review of the supplemental material, it is completely reasonable to request that the authors provide at least one (if not both) of the following in order to enable accessible reproducibility of the presented results:

1) A script that implements a pipeline to carry out the computational workflow in order to generate the models and results in the Figures (notably Table 1 and Fig 2) and 2) GSMM models that were constructed and used in the analyses available in SBML format (or similarly accessible format that would provide utility with current systems biology tools and programs).

We now provide a .zip file (and a README file its content) with the following:

1) EV1 codes – all the codes for running MTA: (a) First step – finding iMAT flux distribution of the source state (using source state gene expression). (b) Second step – running MTA after achieving source state flux distribution.

The medias used (RPMI and DMEM).

The metabolic reconstruction used for this analysis in both SBML and .mat formats.

2) EV2 codes – running MTA from healthy→cancer with the corresponding structure of healthy/cancer paired metabolic gene expression data
3) EV3 codes – running MTA from healthy→adenoma with the corresponding structure of healthy/adenoma paired metabolic gene expression data
4) EV4 codes – codes to generate the figures of genomic properties of FUT9 and the corresponding structures/KM codes

Corresponding Author Name: Noam Auslander, Andrew Freywald, Franco J. Vizeacoumar, Eytan Ruppin

Journal Submitted to: molecular systems biology

Manuscript Number: MSB-17-7739